# A highly sensitive strategy for monitoring real-time proliferation of targeted cell types in vivo

Hiroto Sugawara[1], Junta Imai ®[1] ✉, Junpei Yamamoto[1], Tomohito Izumi[1], Yohei Kawana[1], Akira Endo[1], Masato Kohata[1], Junro Seike[1], Haremaru Kubo[1], Hiroshi Komamura[1], Yuichiro Munakata[2], Yoichiro Asai[1], Shinichiro Hosaka[1], Shojiro Sawada[2], Shinjiro Kodama[1], Kei Takahashi ®[1], Keizo Kaneko[1] & Hideki Katagiri ®[1]

Cell proliferation processes play pivotal roles in timely adaptation to many biological situations. Herein, we establish a highly sensitive and simple strategy by which time-series showing the proliferation of a targeted cell type can be quantitatively monitored in vivo in the same individuals. We generate mice expressing a secreted type of luciferase only in cells producing Cre under the control of the Ki67 promoter. Crossing these with tissue-specific Cre-expressing mice allows us to monitor the proliferation time course of pancreatic β-cells, which are few in number and weakly proliferative, by measuring plasma luciferase activity. Physiological time courses, during obesity development, pregnancy and juvenile growth, as well as diurnal variation, of β-cell proliferation, are clearly detected. Moreover, this strategy can be utilized for highly sensitive ex vivo screening for proliferative factors for targeted cells. Thus, these technologies may contribute to advancements in broad areas of biological and medical research.

To maintain homeostasis, cells proliferate dynamically according to changing endogenous and exogenous conditions. Cell proliferative processes play pivotal roles in adapting to many biological situations, such as tissue repair and augmenting the amount of tissue in response to increased systemic demand. To precisely understand the mechanisms underlying these complex processes, information regarding how cell proliferative responses change over time is important. At present, however, evaluating the quantitative time courses of cell proliferation in vivo requires enormous efforts and resources, because laborious experiments, including the sacrifice of numerous animals and analyses of the samples at many timepoints, must be performed. Despite these arduous efforts, however, differences among individuals often make precise analyses difficult. Accordingly, the development of simpler approaches to monitoring cell proliferation quantitatively, in the same living individuals in vivo, is thus eagerly awaited, in terms of both

upgrading the quality of experiments and sparing experimental resources.

For this purpose, we planned to develop an in vivo monitoring strategy by sampling of a very small amount of blood and focused on a secreted type of luciferase, Gaussia princeps luciferase (Gluc), which is a highly sensitive reporter for quantitative evaluations of cells in vivo, based on measuring its concentrations in blood[1,2]. More than 90% of Gluc produced in cells is reportedly secreted into the circulation[3]. The Gluc reporter system has several advantages over secreted alkaline phosphatase, a commonly used secreted reporter, in terms of the much-reduced assay time and higher sensitivity in vivo[2]. The Gluc reporter system was reported to be used to detect protein secretion rates[4–6] and promoter activities[7,8]. These findings prompted us to apply the Gluc reporter system to detecting cell proliferation status, in combination with a promoter which reflects cellular proliferation.

[1]Department of Metabolism and Diabetes, Tohoku University Graduate School of Medicine, Sendai, Japan. [2]Division of Metabolism and Diabetes, Faculty of Medicine, Tohoku Medical and Pharmaceutical University, Sendai, Japan. ✉e-mail: imai@med.tohoku.ac.jp

Furthermore, the half-life of Gluc is reportedly short, 20 min, in circulating blood[2]. Therefore, we speculated that this system appears to be suitable for real-time monitoring of cell proliferation, quantitatively over time in vivo.

Ki67 is a commonly used cell marker for detecting and quantifying cell proliferation[9–11]. This protein is induced when cells enter the G1-S phase transition of the cell cycle, and its expression continues throughout the G2 and M phases[10–12]. A 1.5 kb proximal promoter of the human Ki67 gene (Ki67p) was recently reported[13]. In Ki67p-GFP expressing human fibrosarcoma cells, pharmacological or cell contact-induced inhibition of proliferation reportedly attenuates GFP expressions, suggesting that Ki67p activity accurately reflects cell proliferation status[13]. In the present study, therefore, to monitor real-time proliferation of a targeted cell type in vivo, we aimed to produce knock-in mice (Ki67p-LSL-Gluc mice) in which the Gluc gene is inductively expressed under Ki67p, when Cre recombinase functions. By crossing them with mice expressing Cre recombinase, we attempted to perform time series monitoring of the proliferation status of targeted cell types in vivo.

Several lines of evidence indicate that terminally differentiated pancreatic β-cells retain significant proliferative capacity in vivo[14–17]. Pancreatic β-cells proliferate in response to increased systemic demands in several situations, such as obesity[18] and pregnancy[19]. We recently reported neuronal signals from the liver, triggered by activation of the hepatic extracellular signal-regulated kinase (ERK) pathway, to enhance compensatory β-cell proliferation during obesity development through a liver-brain-pancreas neuronal relay[20,21]. As impairment of adaptive proliferation processes leads to blood glucose elevation, elucidating the time courses of β-cell proliferation status is critical for understanding the pathophysiology of diabetes mellitus. However, when and how much β-cells proliferate remain unclear, since no tools have been developed for sequentially detecting the proliferative status of β-cells in the same individuals. Therefore, our first goal was to develop a strategy allowing quantitative monitoring of the courses of β-cell proliferation status over time in the same individuals. Since pancreatic β-cells constitute only a small fraction of whole-body cells and have limited proliferative capacity, this strategy, were it to become a reality, would potentially be applicable to detecting the proliferation status of a variety of cell types.

In the present study, we establish a highly sensitive system by which time series of proliferation of targeted cell types in vivo can be monitored in the same individuals, with no need for animal sacrifice. In particular, the sensitivity of this strategy is high enough to detect physiological proliferation of pancreatic β-cells which constitute a very small population and have limited proliferative capacity. Furthermore, employing this strategy, we make a observation indicating that β-cell proliferation differs between light and dark periods in juvenile mice. These innovative technologies may thus contribute to advancements in numerous areas of biological research.

## Results
### Generation of the real-time cell proliferation reporter system
To monitor real-time cell proliferation, we utilized the proximal promoter of the human Ki67 gene (Ki67p)[13]. The proximal promoter of the Ki67p was PCR amplified using human genome (Agilent Technologies, Santa Clara, CA, USA) as reported previously[13] (Supplementary Table 1). We inserted Ki67p and a loxP-chloramphenicol acetyltransferase (CAT)-polyA-loxP (LSL) upstream from the Gluc reporter gene in the pGLuc Basic-1 vector (NanoLight Technologies, Pinetop, AZ, USA)[3]. The CAT-polyA cassette functions as a stop sequence. Accordingly, Gluc expression is expected to be promoted only in cells that are both proliferating and have expressed Cre (Ki67p-LSL-Gluc construct) (Fig. 1A).

Cell lines are expected to actively proliferate. Therefore, to examine whether proliferating cells do indeed express and secrete

Gluc into culture media after Cre recombination, we first transfected the Ki67p-LSL-Gluc construct into a mouse hepatocyte cell line, Hepa1-6, and a mouse β-cell line, MIN6, followed by infection with an adenovirus containing Cre (Cre-adenovirus). Gluc protein and activities were clearly detected in culture medium, while 1.1 and 1.4% of these Gluc levels, respectively, were detected in Hepa1-6 cell lysates, indicating that the vast majority of Gluc produced in the cells was secreted into the culture medium, an observation consistent with a previous report[3] (Supplementary Fig. 1A). Indeed, Gluc activities were clearly detected in the culture media of Cre-adenovirus-infected Hepa1-6 and MIN6 cells, while being barely detectable in LacZ-adenovirus-infected cells (Fig. 1B, C). Magnitudes of the Gluc activities in media varied between Hepa1-6 and MIN-6 cells. This likely reflects the higher cell proliferation activity in Hepa1-6 than in MIN-6 cells, given that not only expression of the *MKi67* gene, but also ratios of Ki-67 or phospho-histone H3 (PHH3) protein-positive cells were significantly higher in Hepa1-6 than in MIN-6 cells (Supplementary Fig. 1B–D).

Next, we treated Hepa1-6 cells with cell cycle inhibitors and evaluated cell proliferative status. Increased Gluc activity in culture media was significantly decreased by treating Hepa1-6 cells with cell cycle inhibitors, such as mitomycin C (Fig. 1D) and rapamycin (Fig. 1E), or a CDK inhibitor, roscovitine (Fig. 1F) for 24 h. Gluc activities in culture media began to decrease from 8 h after starting the cell cycle inhibitor treatments, and thereafter almost complete inhibition of the activities persisted from 24 to 96 h after the start of these treatments (Supplementary Fig. 2A–C). Consistently, *MKi67* gene expressions in Hepa1-6 cells were markedly decreased by 24-h treatment with the cell cycle inhibitors (Supplementary Fig. 2D–F). In addition, *MKi67* gene expressions of cells and Gluc concentrations in media showed a strong and significant correlation ($r = 0.903$, $P < 0.0001$) (Supplementary Fig. 2G). These findings indicate that Gluc activity in the culture media reflects the cell proliferation status of these cells. The quantitative reverse transcriptase-PCR (qRT-PCR) procedure showed that gene expressions of the loxP-flanked CAT gene and the Gluc gene were significantly downregulated and upregulated, respectively, in MIN-6 cells after the Cre-adenovirus infection (Supplementary Fig. 3A, B), indicating Cre to mediate recombination of the transgene cassette as intended. Collectively, these results show that the Ki67p-LSL-Gluc system allows us to monitor the cell proliferation status in cultured cell lines after Cre recombination by measuring the activity of the luciferase secreted into culture media.

### In vivo real-time monitoring of hepatocyte proliferation after partial hepatectomy
Next, we knocked the Ki67p-LSL-Gluc construct into the Rosa26 locus in C57BL/6 N mice (Ki67p-LSL-Gluc mice). In these mice, Gluc was expected to be expressed only in the Cre-functioning cells under control of the Ki67 promoter and to be secreted into the circulation. The liver is a large organ with high proliferative capacity[22–24]. Therefore, we speculated that the liver would be an ideal organ for assessing the feasibility of this system for monitoring cell proliferation in vivo. Accordingly, we first generated tamoxifen-inducible liver-specific Ki67p-Gluc mice (iLKi67p-Gluc mice) by crossing albumin-Cre-ER mice[25] with Ki67p-LSL-Gluc mice.

Partial hepatectomy (PHx) is commonly used to induce hepatocyte proliferation[23,24]. Therefore, we attempted to monitor in vivo real-time hepatocyte proliferation of iLKi67p-Gluc mice which had undergone PHx. A total of 20 μL of plasma obtained from the tail vein were applied for assaying the luciferase activity at each timepoint. The luciferase activity in plasma started to increase between days 1 and 2, and the peak elevation was observed on day 2 after PHx (Fig. 2A). Luciferase activity had fallen, by day 5, to levels only slightly higher than those on day 0, indicating that marked hepatocyte proliferation had occurred acutely after the PHx. Importantly, consistent with

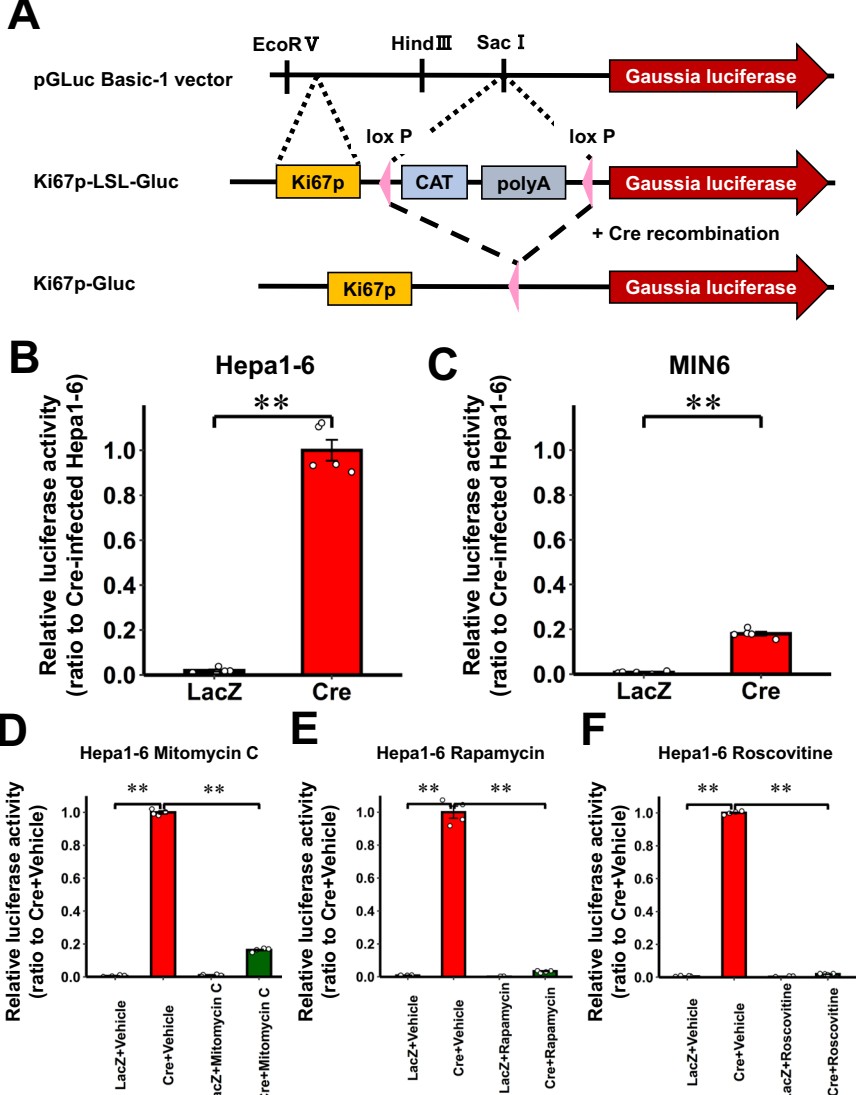

**Fig. 1 | Generation of the real-time cell proliferation reporter system.**
**A** Construct of the real-time cell proliferation reporter system. The CAT/polyA element flanked by loxP sites (LSL cassette) was located downstream from the human Ki67 promoter (Ki67p) in the pGLuc Basic-1 vector and comprised the Ki67p-LSL-Gluc construct. After Cre recombination, the LSL cassette was derived and Gluc was expressed under Ki67p activity. **B**, **C** Luciferase activity in culture media of LacZ- or Cre-adenovirus infected Hepa1-6 cells **B** or MIN6 **C** cells relative to that in culture media of Cre-adenovirus infected Hepa1-6 cells (Cre-infected Hepa1-6). **D**–**F** Luciferase activity in culture media of LacZ- or Cre-adenovirus infected Hepa1-6 cells, after treatment with Mitomycin C **D**, Rapamycin **E**, or Roscovitine

**F** relative to those in culture media of Cre-adenovirus infected cells treated with vehicle. Data are presented as means ± SEM. **\*\*p < 0.01, assessed by two-sided unpaired t-test **B**, **C**, or one-way ANOVA followed by Bonferroni's post hoc test **D**–**F**. **B**, **C** n = 5 independent samples for each group. **D**–**F** n = 4 independent samples for each group. Results are representative of two independent experiments. Exact P values are **B** P = 2.9E-8; **C** P = 5.0E-8; **D** P = 2.6E-19 (LacZ + Vehicle vs. Cre + Vehicle), P = 2.1E-18 (Cre + Vehicle vs. Cre + Mitomycin C); **E** P = 4.1E-13 (LacZ + Vehicle vs. Cre + Vehicle), P = 5.6E-13 (Cre + Vehicle vs. Cre + Rapamycin); **F** P = 2.2E-24 (LacZ + Vehicle vs. Cre + Vehicle), P = 2.6E-24 (Cre + Vehicle vs. Cre + Roscovitine). Source data are provided as a Source Data file.

results of plasma luciferase activity (Fig. 2A), proportions of hepatocytes positive for one of two distinct cell proliferation markers, Ki67 and PHH3, were both markedly increased on day 2 after PHx, but had returned to the basal level by day 9 after PHx (Fig. 2B and Supplementary Fig. 4A). In addition, proportions of either Ki67- or PHH3-positive hepatocytes and plasma Gluc activities showed a marked and significant correlation in the mice used in these experiments ($r = 0.904$, $P < 0.0001$ and $r = 0.974$, $P < 0.0000001$, respectively) (Fig. 2C and Supplementary Fig. 4B). Furthermore, time courses of the luciferase activity in iLKi67p-Gluc mice after PHx were similar to the previously reported time courses of murine liver regeneration after PHx, peaking between 36 and 48 h after the operation[23,24]. The luciferase activity in plasma of iLKi67p-Gluc mice after sham operation

remained at very low, essentially basal, levels throughout the experimental period (Fig. 2A).

We next performed bioluminescent analysis. After venous injection of coelenterazine, a substrate of luciferase, strong luminescence signals were detected at the site of the remnant liver, only in iLKi67p-Gluc mice which had undergone PHx (Fig. 2D). No luminescence was detected in the two types of negative control, Ki67p-LSL-Gluc mice without the Cre-ER transgene which had undergone PHx, and iLKi67p-Gluc mice which had undergone sham operation (Fig. 2D). In addition, luminescence signals were significantly increased at the site of the remnant liver on day 2 compared to those before PHx, and had reverted to the basal levels on day 9 after the PHx (Fig. 2E), which is consistent with the results of plasma Gluc

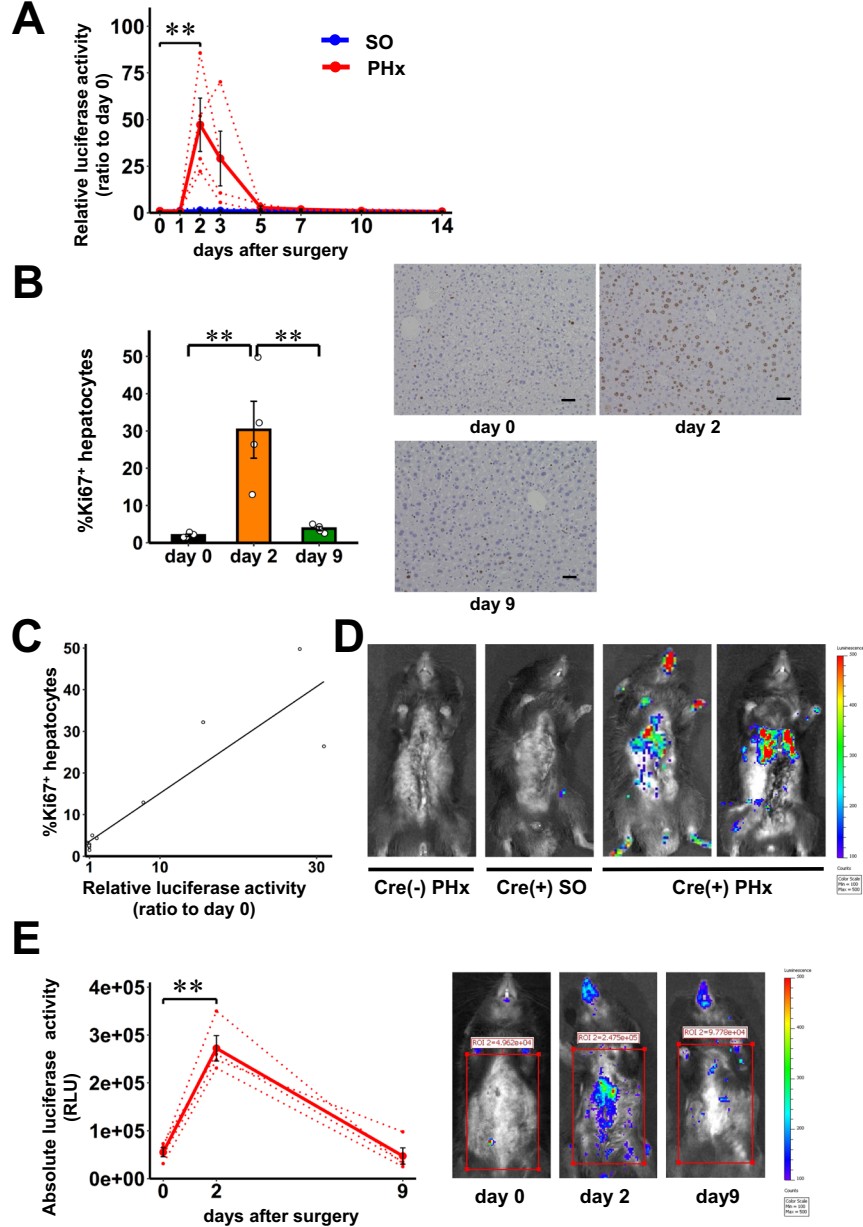

**Fig. 2 | In vivo real-time monitoring of hepatocyte proliferation after partial hepatectomy. A** Time courses of luciferase activity in plasma of 10 weeks old male iLKi67p-Gluc mice on C57BL/6 background after partial hepatectomy (PHx, red) relative to those on day 0. 10 weeks old male iLKi67p-Gluc mice undergoing sham operation (SO, blue) served as controls. Solid lines and dotted lines indicate average and individual values, respectively. **B** Ki67-positive hepatocyte ratios in 10 weeks old male iLKi67p-Gluc mice on day 0, 2, or 9 after PHx; representative images are shown in the right three panels. Scale bars denote 50 μm. **C** Linear relationship between relative luciferase activity and Ki67-positive hepatocyte ratios. Open circles indicate relative luciferase activity and Ki67-positive hepatocytes in individual 10 weeks old male iLKi67p-Gluc mice after PHx relative to those on day 0 (r = 0.904, P = 5.3E-5). **D** In vivo bioluminescence imaging of 10 weeks old male iLKi67p-Gluc mice on day 3 after PHx (Cre (+) PHx). Ten weeks old male Ki67p-LSL-Gluc mice without the Cre-ER transgene which had undergone PHx (Cre (-)

PHx), and iLKi67p-Gluc mice which had undergone sham operation (Cre (+) SO) served as controls. Representative images are shown. (E) Quantification of signal intensity in the PHx group on 0, 2, and 9 days after PHx; representative bioluminescence images are shown. Rectangles indicate regions of interest. Solid lines and dotted lines indicate average and individual values, respectively. Data are presented as means ± SEM. **p < 0.01, assessed by one-way repeated-measures ANOVA followed by the Tukey multiple comparison test for the various time points **A**, **E**, or one-way ANOVA followed by Bonferroni's post hoc test **B**. Pearson's correlation coefficient (two-sided) was used to determine the correlation **C**. **A** n = 4 independent animals for each group, from two independent experiments. **B**, **C** n = 4 independent animals for each group, from three independent experiments. (E) n = 4 independent animals, from three independent experiments. Exact P values are **A** P = 0.0027; **B** P = 0.0043 (day 0 vs. day 2), P = 0.0065 (day 2 vs. day 9); **E** P = 4.5E-4 (day 0 vs. day 2). Source data are provided as a Source Data file.

activity (Fig. 2A). Therefore, luciferase activity elevation in plasma is likely to be attributable to selective enhancement of Gluc production in the liver after PHx. Thus, these results indicate the availability of the Ki67p-Gluc system for quantitative real-time in vivo monitoring of the proliferation status of selected cell types by sampling of a very small amount of blood.

## In vivo real-time monitoring of β-cell proliferation induced by neuronal signaling

Next, we explored whether the proliferation of pancreatic β-cells can be monitored in vivo employing the Ki67p-Gluc system. We crossed RIP-Cre-ER mice[14] with Ki67p-LSL-Gluc mice, thereby generating tamoxifen-inducible β-cell-specific Ki67p-Gluc mice (iβKi67p-Gluc

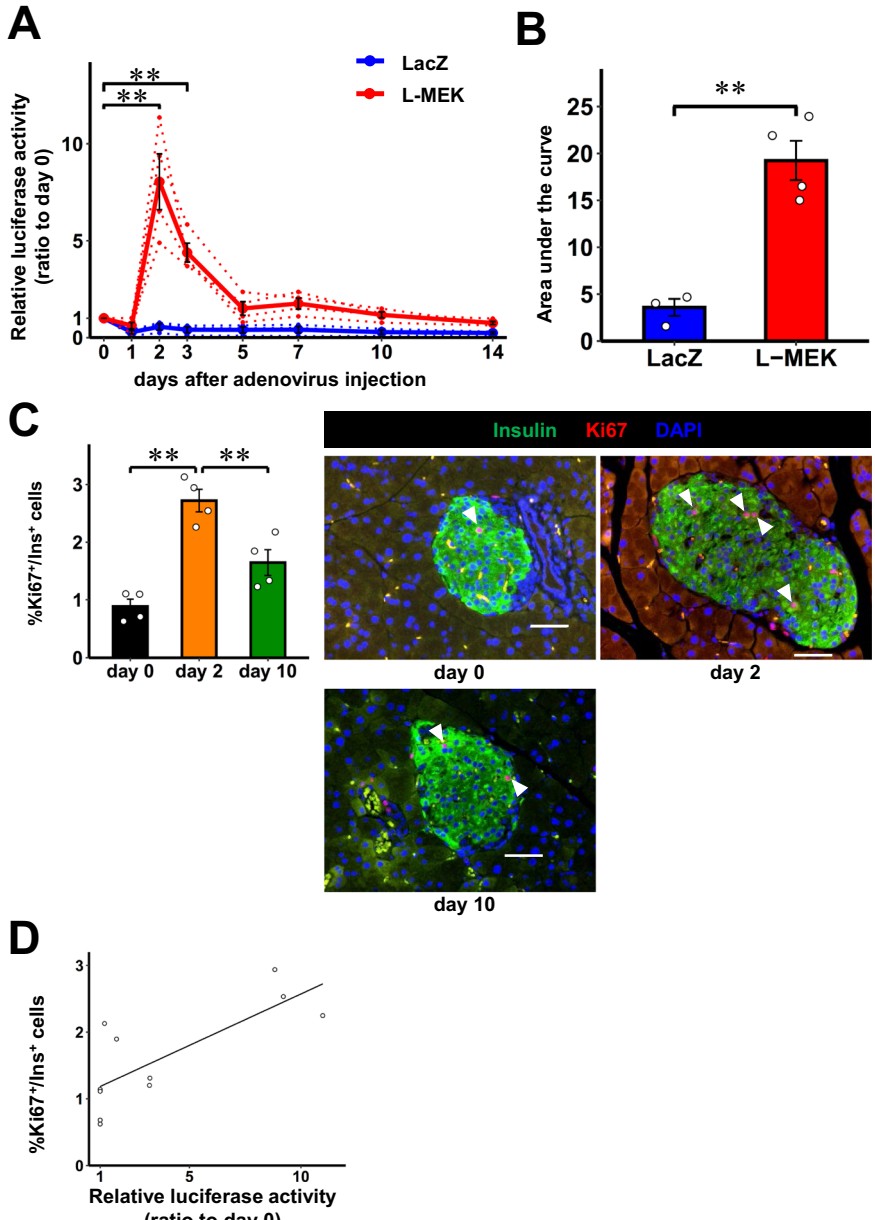

**Fig. 3 | In vivo real-time monitoring of β-cell proliferation induced by neuronal signaling. A** Time courses of luciferase activity in plasma from 3 months old male iβKi67p-Gluc mice on C57BL/6 background after L-MEK administration relative to those on day 0 (L-MEK, red). 3 months old male iβKi67p-Gluc mice after LacZ administration served as controls (LacZ, blue). Solid lines and dotted lines indicate average and individual values, respectively. **B** The area under the curve of the plasma luciferase activity of L-MEK or LacZ during the experimental period. **C** Ki67 and insulin co-positive cell (Ki67⁺/Ins⁺ cells) ratios in insulin positive cells of 3 months old male iβKi67p-Gluc mice on day 0, 2, and 10 after L-MEK administration relative to those on day 0; representative images are shown in the right three panels. Each arrowhead denotes a Ki67⁺/Ins⁺ cell. Scale bars denote 50 μm. **D** Linear relationship between relative luciferase activity and %Ki67⁺/Ins⁺ β-cells. Open circles indicate relative luciferase activity and %Ki67⁺/Ins⁺ β-cells in individual iβKi67p-Gluc mice after L-MEK administration relative to those on day 0 ($r = 0.835$, $P = 0.00073$). Data are presented as means ± SEM. **$p < 0.01$, assessed by one-way repeated-measures ANOVA followed by Tukey multiple comparison test for the various time points **A**, two-sided unpaired t-test **B**, or one-way ANOVA followed by Bonferroni's post hoc test **C**. Pearson's correlation coefficient (two-sided) was used to determine the correlation **D**. **A**, **B** $n = 4$ independent samples for L-MEK group, n = 3 independent samples for LacZ group. Results are representative of two independent experiments. **C**, **D** $n = 4$ independent animals for each group, from four independent experiments. Exact $P$ values are **A**, $P = 2.4E-7$ (day 0 vs. day 2), $P = 0.0046$ (day 0 vs. day 3); **B** $P = 0.0018$; **C** $P = 1.9E-4$ (day 0 vs. day 2), P = 0.0082 (day 2 vs. day 10). Source data are provided as a Source Data file.

mice). We reported that a neuronal relay consisting of the splanchnic nerve from the liver and vagal nerve to the pancreas mediates large amounts of β-cell proliferation[20,21]. This neuronal relay is triggered by activation of the hepatic ERK pathway[20,21]. In these prior studies, the hepatic ERK pathway was activated by recombinant adenoviral expression of the active mutant gene of mitogen-activated protein kinase/ERK kinase1 (MEK-1)[20,21]. Therefore, to examine whether the Ki67p-Gluc strategy is useful for monitoring the proliferation of β-cells,

a small population among whole-body cells, we administered recombinant adenovirus containing the active mutant gene of MEK-1 in the livers of iβKi67p-Gluc mice (L-MEK mice), followed by monitoring luciferase activity in plasma samples collected from these mice. We used iβKi67p-Gluc mice treated with LacZ adenovirus (LacZ mice) as controls. Strikingly, the luciferase activity in plasma from L-MEK mice was markedly upregulated as compared to that of LacZ mice (Fig. 3A). The peak elevation of the luciferase activity in plasma of L-MEK mice

was seen on day 2 after adenoviral administration (Fig. 3A). The luciferase activity decreased to one quarter of the peak level on day 5, followed by further gradual decreases until day 14 (Fig. 3A). In contrast, the luciferase activity in plasma of LacZ mice remained low throughout the experimental period (Fig. 3A). The area under the curve (AUC) of the plasma luciferase activity in L-MEK mice during the experimental period was significantly greater than that in LacZ mice (Fig. 3B). Importantly, the time courses of the luciferase activity in L-MEK mice were similar to those of bromodeoxyuridine (BrdU) incorporation into β-cells after adenoviral ERK activation in the liver, as described in our previous report[20]. Consistently, ratios of either Ki67- or PHH3-positive β-cells in L-MEK mice were markedly increased on day 2 after PHx followed by significant decrements on day 10 after adenoviral administration (Fig. 3C and Supplementary Fig. 5A). In addition, ratios of Ki67- and PHH3-positive β-cells showed marked and significant correlation with plasma Gluc activities in the mice used in these experiments ($r = 0.835$, $P < 0.001$ and $r = 0.823$, $P < 0.005$, respectively) (Fig. 3D and Supplementary Fig. 5B). Collectively, this system is sufficiently sensitive to detect pancreatic β-cell proliferation despite these cells constituting only a very small population at the whole-body level.

Since Gluc and insulin are both secreted proteins, insulin and Gluc secretions might impact each other. To assess this possibility, we first examined high glucose (HG)- and KCl- stimulated insulin secretions of islets isolated from iβKi67pGluc mice. Both HG and KCl significantly stimulated insulin secretions from iβKi67pGluc mouse islets, and insulin secretory responses were comparable to those from RIP-Cre- and floxed mice (Supplementary Fig. 6A). In addition, insulin contents of isolated islets were similar in iβKi67pGluc, RIP-Cre and floxed mice (Supplementary Fig. 6B). Thus, β-cells in iβKi67p-Gluc are likely to secrete insulin normally. We next performed glucose tolerance tests in L-MEK mice on day 2 after adenoviral administration, when Gluc expression in β-cells was markedly upregulated, followed by monitoring both the insulin level and luciferase activity in plasma samples collected from these mice. Thirty minutes after the glucose administrations, both blood glucose and plasma insulin were significantly increased as compared to pre-glucose loading levels (Supplementary Fig. 7A, B). In contrast, plasma luciferase activity was not altered after glucose loading (Supplementary Fig. 7C). Thus, insulin secretion is unlikely to have an impact on the Gluc secretory process in β-cells. Taken together, these findings indicate that our system is feasible for in vivo real-time monitoring of β-cell proliferation as it minimally affects, while itself being minimally affected by, insulin secretion.

### In vivo real-time monitoring of physiological β-cell proliferation

We, thereafter, attempted to monitor physiological β-cell proliferation. Pancreatic β-cell proliferation is known to be physiologically promoted under certain conditions, such as pregnancy[26,27] and obesity development[18]. Therefore, to evaluate the feasibility of our strategy for monitoring physiological β-cell proliferation status, we continually measured luciferase activity in plasma samples from pregnant iβKi67p-Gluc mice. Mated female mice, which subsequently delivered, were defined as having been pregnant, while unmated mice were used as controls. Luciferase activity in the plasma obtained from pregnant iβKi67p-Gluc mice started to increase on day 3 after mating, and the elevations persisted until day 12 of pregnancy, followed by gradual decrements until delivery (Fig. 4A). The AUC of the plasma luciferase activity of pregnant iβKi67p-Gluc mice during the experimental period was significantly greater than that of the control mice (Fig. 4B). Consistent with the results of plasma luciferase activity, the proportions of Ki67- and PHH3-positive β-cells in pregnant iβKi67p-Gluc mice were significantly increased on day 12, followed by significant decrements on day 21, of pregnancy (Fig. 4C and Supplementary Fig. 8A). In addition, proportions of Ki67- and PHH3-positive β-cells both showed

strong and significant correlations with plasma Gluc activities ($r = 0.812$, $P < 0.005$ and $r = 0.777$, $P < 0.005$, respectively) in the mice used in these experiments (Fig. 4D and Supplementary Fig. 8B). Again, time courses of the luciferase activity in pregnant iβKi67p-Gluc mice were similar to those previously reported during pregnancy[26,28,29]. These results clearly showed that the physiological time course of β-cell proliferation during pregnancy can also be monitored employing the Ki67p-Gluc system.

To further examine whether this system is able to monitor low-level and continuous β-cell proliferation under conditions of obesity, iβKi67p-Gluc mice were loaded with a high fat diet (HFD). The luciferase activity in the plasma of HFD-fed iβKi67p-Gluc mice was slightly increased from one week after starting the HFD loading, and sustained throughout the experimental period, for 8 weeks (Fig. 5A). Meanwhile, the luciferase activity in the plasma of normal chow-fed iβKi67p-Gluc mice remained at the basal level throughout the experimental period (Fig. 5A). Notably, the AUC of the plasma luciferase activity in HFD-fed iβKi67p-Gluc mice during the experimental period was significantly greater than that in the control mice (Fig. 5B). Consistent with the results of plasma luciferase activity, the proportions of Ki67- and PHH3-positive β-cells in HFD-fed iβKi67p-Gluc mice were significantly increased 8 weeks after starting the HFD (Fig. 5C and Supplementary Fig. 9A). In addition, the proportions of Ki67- and PHH3-positive β-cells both showed a significant correlation with plasma Gluc activities ($r = 0.891$, $P < 0.005$ and $r = 0.694$, $P < 0.05$ respectively) in the mice used for these experiments (Fig. 5D and Supplementary Fig. 9B). Thus, low levels of continuous and physiological β-cell proliferation induced by HFD can be monitored employing this system. In particular, repetitive and easy collections of samples over time from the same individuals enable us to detect low-grade but continuous proliferation by evaluating the AUC of plasma luciferase activity.

The β-cell secretory pathway is reportedly affected by endoplasmic reticulum (ER) stress[30,31] induced under conditions such as obesity and diabetes. To examine whether increased ER stress affects the Gluc secretory process, we treated islets isolated from iβKi67p-Gluc mice under the HG condition, which reportedly promotes β-cell proliferation[32,33] and also induces ER stress in β-cells[34,35]. Expressions of *Bip*, an ER stress marker, were dose-dependently and significantly increased along with elevation of glucose concentrations in culture media (Supplementary Fig. 10A), indicating that ER stress is indeed induced in isolated islets by HG. Under these experimental conditions, Gluc activities in culture media were also significantly increased according to elevation of glucose concentrations in culture media (Supplementary Fig. 10B). Furthermore, *Mki67* expressions of islet cells and Gluc activities in culture media at these glucose concentrations, 5.5, 15 and 25 mM, showed a strong correlation ($r = 0.766$, $P < 0.001$) (Supplementary Fig. 10C). Thus, effects of ER stress were minimal on Gluc secretion from β-cells in iβKi67p-Gluc mice.

Taken together, our observations indicate that this system allows us to continually detect proliferation of very small populations of cells in vivo, probably due to the high sensitivity of Gluc as well as the high selectivity and low background level of Ki67p.

### In vivo monitoring of β-cell proliferation during the juvenile period and diurnal variation of β-cell proliferation

The success in monitoring physiological β-cell proliferation prompted us to apply this system to assessing the physiological time course of this proliferation, which has been difficult to elucidate employing pre-existing methods.

β-cell proliferation is reportedly high in the initial months of life[36-39]. However, the natural time course of β-cell proliferation in the same individuals during the juvenile period is unknown due to the lack of reliable tools for evaluation. To tackle this issue, we monitored β-cell proliferation during the juvenile period by continually collecting 10 μL plasma samples from mice every week starting at 4 weeks of age. In

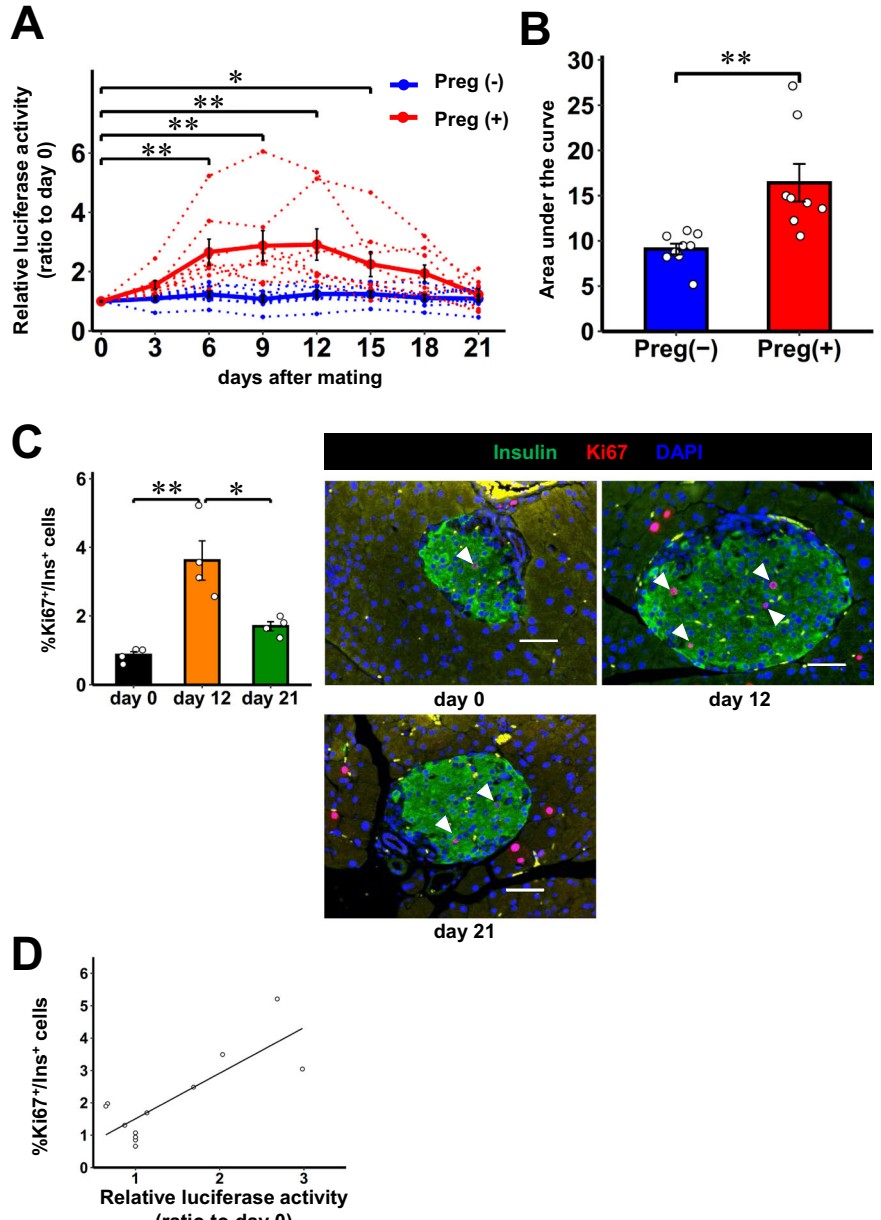

**Fig. 4 | In vivo real-time monitoring of β-cell proliferation during pregnancy.**
**A** Time courses of luciferase activity in plasma of 3 months old female iβKi67p-Gluc mice on C57BL/6 background during pregnancy relative to those on day 0. Preg (+) indicates mice which had become pregnant, based on subsequent delivery (red). Preg (-) indicates mice which had not been mated. The day when a vaginal plug was identified was defined as day 0 (blue). Solid lines and dotted lines indicate average and individual values, respectively. **B** The area under the curve of the plasma luciferase activity of Preg (+) mice or Preg (-) mice during the experimental period. **C** Ki67+/Ins+ cell ratios in insulin positive cells of 3 months old female iβKi67p-Gluc mice on day 0, 12, and 21 after mating relative to those on day 0; representative images are shown in the right three panels. Each arrowhead denotes a Ki67+/Ins+ cell. Scale bars denote 50 μm. **D** Linear relationship between relative luciferase activity and %Ki67+/Ins+ β-cells. Open circles indicate relative luciferase activity and %Ki67+/Ins+ cells in individual iβKi67p-Gluc mice after mating relative to those on day 0 ($r = 0.812$, $P = 0.0013$). Data are presented as means ± SEM. *$p < 0.05$, **$p < 0.01$, assessed by one-way repeated-measures ANOVA followed by Tukey multiple comparison test for the various time points **A**, two-sided unpaired t-test **B**, or one-way ANOVA followed by Bonferroni's post hoc test **C**. Pearson's correlation coefficient (two-sided) was used to determine the correlation **D**. **A**, **B** $n = 8$ independent animals for Preg (+), $n = 9$ independent animals for Preg (-), from ten independent experiments. **C**, **D** $n = 4$ independent animals for each group, from three independent experiments. Exact P values are **A**, $P = 0.0013$ (day 0 vs. day 6), $P = 2.0E-4$ (day 0 vs. day 9), $P = 1.4E-4$ (day 0 vs. day 12), $P = 0.032$ (day 0 vs. day 15); B, $P = 0.0087$; **C** $P = 9.5E-4$ (day 0 vs. day 12), $P = 0.010$ (day 12 vs. day 21). Source data are provided as a Source Data file.

these experiments, we use congenital β-cell-specific Ki67p-Gluc mice (cβKi67p-Gluc mice), generated by crossing RIP-Cre mice with Ki67p-LSL-Gluc mice, because tamoxifen administration is difficult in neonatal mice due to their bodies being very small. Luciferase activity in the plasma obtained from cβKi67p-Gluc mice was maintained from 4 to 5 weeks of age, and then decreased gradually until 7 weeks of age. Thereafter, similar levels persisted until 12 weeks of age (Fig. 6A).

Notably, luciferase activity in the plasma obtained from 4-5-week-old mice was significantly higher, by 2.8-fold, than that in plasma samples from 12-week-old mice. To confirm the validity of these results, β-cell proliferation was examined by conventional gene expression analyses of *MKi67* in isolated islet cells. Consistent with the results obtained from experiments using cβKi67p-Gluc mice, gene expressions of *MKi67* in islet cells from 4-week-old mice were significantly higher than

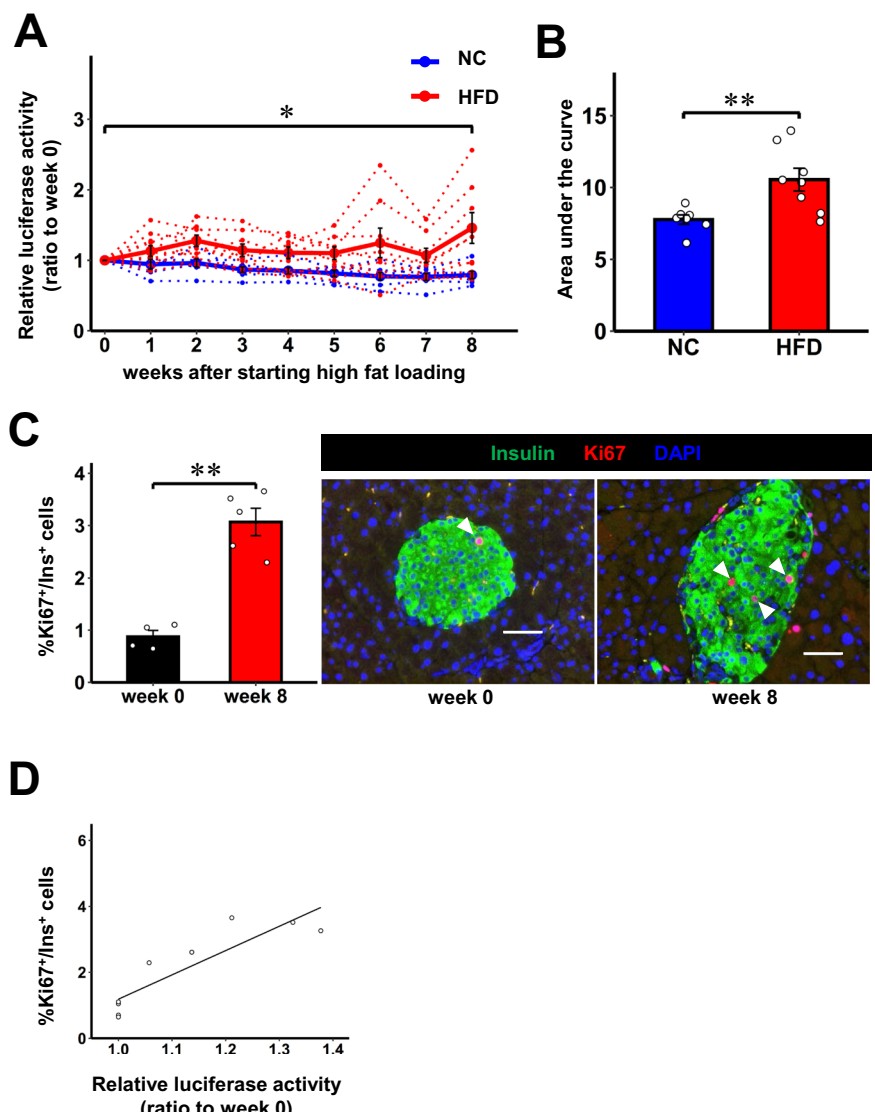

**Fig. 5 | In vivo real-time monitoring of β-cell proliferation during high fat loading.** **A** Time courses of luciferase activity in plasma of 3 months old male iβKi67p-Gluc mice on C57BL/6 background fed high fat-diet (HFD, red) relative to those on week 0. iβKi67p-Gluc mice which fed normal chow (NC, blue) served as controls. Solid lines and dotted lines indicate average and individual values, respectively. **B** The area under the curve of the plasma luciferase activity of HFD or NC during the experimental period. **C** Ki67⁺/Ins⁺ cell ratios in insulin positive cells of 3 months old male iβKi67p-Gluc mice on week 0 and 8 after high fat loading relative to those on week 0; representative images are shown in the right two panels. Each arrowhead denotes a Ki67⁺/Ins⁺ cell. Scale bars denote 50 μm. **D** Linear relationship between relative luciferase activity and %Ki67⁺/Ins⁺ β-cells. Open circles indicate relative luciferase activity and %Ki67⁺/Ins⁺ β-cells in individual iβKi67p-Gluc mice after high fat loading relative to those on week 0 ($r = 0.891$, $P = 0.0013$). Data are presented as means ± SEM. **A, B** *$p < 0.05$, **$p < 0.01$, assessed by one-way repeated-measures ANOVA followed by Tukey multiple comparison test for the various time points **A**, two-sided unpaired t-test **B**, or two-sided paired t-test **C**. Pearson's correlation coefficient (two-sided) was used to determine the correlation **D**. $n = 8$ independent animals for HFD, n = 7 independent animals for NC, from five independent experiments. **C, D** $n = 5$ independent animals for HFD, n = 4 independent animals for NC, from three independent experiments. Exact P values are **A** $P = 0.024$ (week 0 vs. week 8); **B** $P = 0.0018$; **C** $P = 0.00023$. Source data are provided as a Source Data file.

those from 8-week-old mice (Supplementary Fig. 11A). These findings clarify that β-cell proliferation during the juvenile period promptly decreases starting at 5 weeks of age, and is minimal by about 7 weeks after birth.

Circadian control of cell proliferation was reported in several types of cells[40]. However, whether there is diurnal variation of physiological β-cell proliferation in vivo remains unknown. Therefore, we measured luciferase activity in the plasma of 4-week-old cβKi67p-Gluc mice repeatedly during a single day to explore whether β-cell proliferation shows diurnal variation. We collected 10 μL samples of plasma from these mice at 4-h intervals. The luciferase activity was maintained at high levels during the light period and decreased prior to the beginning of the dark period. Meanwhile, the luciferase activity reached a nadir at the initiation of the dark period and then gradually increased toward the start of the light period (Fig. 6B). The AUC of the plasma luciferase activity during the light period was significantly greater than that during the dark period (Fig. 6C). Again, we confirmed that the β-cell proliferation at ZT16 was significantly lower than that at ZT0 by conventional gene expression analyses of *MKi67* in isolated islet cells (Supplementary Fig. 11B), observations consistent with the results obtained from experiments using cβKi67p-Gluc mice (Fig. 6B). These results indicate successful detection of diurnal variation in β-cell proliferation. Thus, our system allows real-time monitoring of the proliferation of targeted cell types in vivo, in the same living mice, not only daily and weekly, but also on the diurnal time axis.

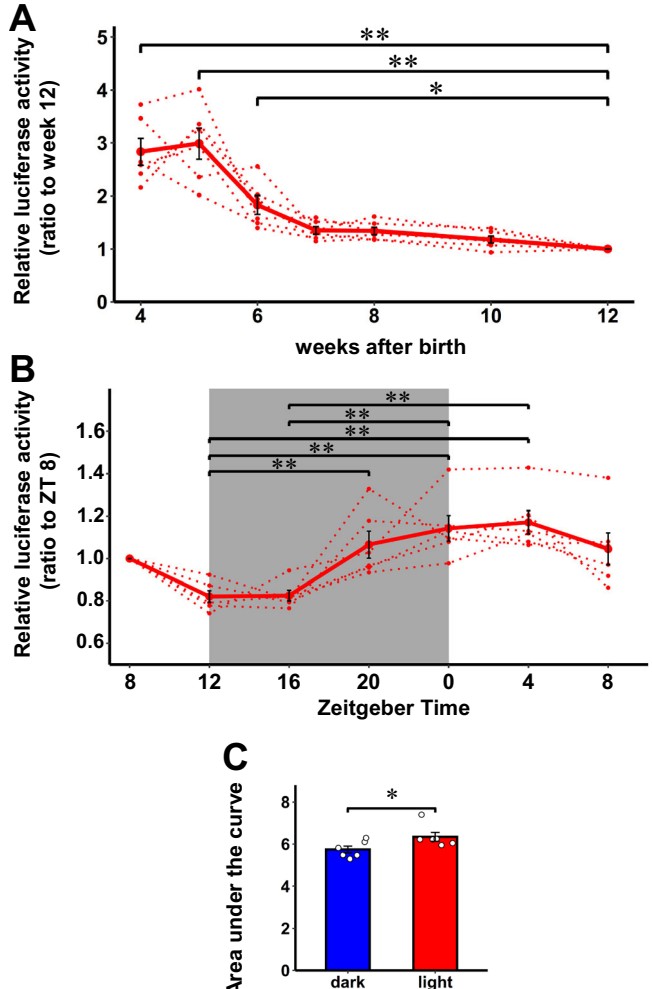

**Fig. 6 | In vivo real-time monitoring of β-cell proliferation in the juvenile stage.** **A** Time courses of luciferase activity in plasma from male cβKi67p-Gluc mice on C57BL/6 background relative to those on week 12. Solid lines and dotted lines indicate average and individual values, respectively. **B** The diurnal variation in luciferase activity in plasma from 4 to 5 weeks-old male cβKi67p-Gluc mice. Luciferase activity relative to that on ZT 8 is shown. Solid lines and dotted lines indicate average and individual values, respectively. **C** The area under the curve of the plasma luciferase activity of 4 weeks-old male cβKi67p-Gluc mice during the dark (ZT12-24) or light period (0–12). Data are presented as means ± SEM. *$p < 0.05$, **$p < 0.01$, assessed by one-way repeated-measures ANOVA followed by the Tukey multiple comparison test for the various time points **A**, **B**, or two-sided paired t-test **C**. **A** $n = 6$ independent samples. Results are representative of two independent experiments. **B**, **C** $n = 6$ independent samples from two independent experiments. Exact P values are A, $P = 9.4E-8$ (4 weeks vs. 12 weeks), $P = 1.7E-8$ (5 weeks vs. 12 weeks), $P = 0.0139$ (6 weeks vs. 12 weeks); **B** $P = 0.0086$ (ZT 12 vs. ZT 20), $P = 3.3E-4$ (ZT 12 vs. ZT 0), $P = 9.5E-5$ (ZT 12 vs. ZT 4), $P = 4.0E-4$ (ZT 16 vs. ZT 0), $P = 1.1E-4$ (ZT 16 vs. ZT 4); **C** $P = 0.021$. Source data are provided as a Source Data file.

### Ex vivo search for β-cell trophic factors using isolated islets from iβKi67p-Gluc mice

The β-cell amount is a crucial determinant of glucose homeostasis, and increasing the β-cell mass is a long-awaited therapeutic strategy for treating diabetes mellitus[41–43], although as yet there are no clinically applicable strategies for increasing β-cell mass. As described above (Supplementary Fig. 10C), luciferase activities in the culture media at several glucose concentrations correlated strongly with *MKi67* expressions. Thus, the luciferase activity elevation in the culture media appears to reflect β-cell proliferation, and the Ki67p-Gluc system is speculated to be applicable to searching for β-cell trophic factors. We next cultured islets isolated from iβKi67p-Gluc mice with glucagon-like

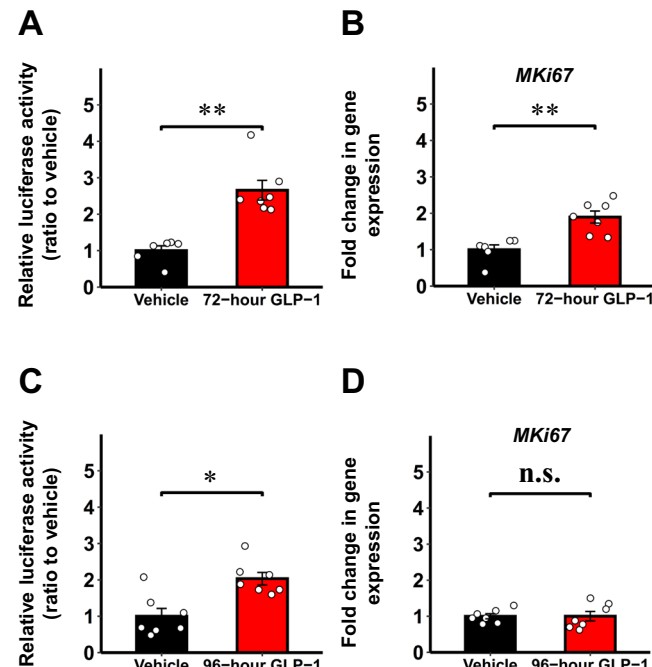

**Fig. 7 | Ex vivo monitoring of β-cell proliferation using isolated islets from iβKi67p-Gluc mice.** **A** Luciferase activity in culture media of isolated islets from 3 months old male iβKi67p-Gluc mice on C57BL/6 background after stimulation with glucagon-like peptide-1 (GLP-1) for 72 h (72-h GLP-1). **B** Gene expression of mouse Ki67 in these islets. **C** Luciferase activity in culture media of isolated islets from 3 months old male iβKi67p-Gluc mice after stimulation with GLP-1 for 96 hours (96-h GLP-1). **D** Gene expression of mouse Ki67 in these islets. Luciferase activity in culture media of isolated islets cultured with vehicle or gene expression in these islets served as control (Vehicle). Data are presented as means ± SEM. n.s., not significant, *$p < 0.05$, **$p < 0.01$, assessed by two-sided unpaired t-test. **A**, **B** $n = 7$ independent samples for 72-h GLP-1, $n = 6$ independent samples for Vehicle. **C**, **D** $n = 7$ independent samples for each group. Exact P values are **A**, $P = 9.4E-8$ (4 weeks vs. 12 weeks), $P = 1.7E-8$ (5 weeks vs. 12 weeks), $P = 0.00029$; **B** $P = 0.0017$; **C** $P = 0.0028$. Source data are provided as a Source Data file.

peptide-1 (GLP-1), another β-cell proliferative stimulus[33,44], followed by estimating the luciferase activity in the culture media. After 72-h GLP-1 stimulation, both the luciferase activity (Fig. 7A) in the culture media and gene expressions of *MKi67* in the islet cells (Fig. 7B) were significantly increased. Thus, our approach allows the detection of trophic factor-mediated β-cell proliferation by employing a simple method, i.e., collecting the culture media, with sensitivity similar to that of gene expression analysis.

Interestingly, after 96-hour stimulation with GLP-1, significant elevations of luciferase activity in culture media (Fig. 7C), but not *MKi67* gene expressions (Fig. 7D), were detected. Since Gluc is reported to be very stable with a half-life of approximately 6 days in culture media[1], in contrast to its rapid clearance from blood[2], this method may allow us to cumulatively estimate proliferative status throughout the stimulation period, even after the proliferation itself has ended. This feature of Gluc may achieve sensitive detection of cell proliferation during broader time windows with stimulation, while gene expression analyses reflect cell proliferation status only at the timepoint of cell sample collection.

## Discussion

In the present study, we established a strategy by which time series showing proliferation of targeted cell types in vivo can be quantitatively monitored in the same living mouse employing a simple blood analysis approach. Notably, our results clearly show that in vivo real-time proliferation of pancreatic β-cells, a very small cell population at

the whole-body level, can be monitored employing this system. A previous report showed pancreatic islet cells to comprise only 1–4% of all pancreatic cells[45], and the β-cell population reportedly accounts for 80% of rodent islet cells[46]. Moreover, β-cells exhibit a very low rate of proliferation in the steady state[39] and are resistant to proliferation even in response to several stimuli. Indeed, only approximately 3 and 1% of all β-cells are reportedly BrdU-positive during pregnancy and obesity development[47,48], respectively. Thus, proliferating β-cells even under these conditions were quite rare. However, in this study, a sharp rise in plasma luciferase activity was observed in L-MEK mice (Fig. 3), in which BrdU-positive cells were reported to be relatively abundant (6% of β-cells) on day 3 after MEK adenovirus administration[20]. Prompt reduction of BrdU-positive β-cells on day 9 after MEK adenovirus administration[20] was also replicated by this strategy. In addition, mild but sustained elevation of plasma luciferase activity was observed in both pregnant (Fig. 4) and HFD-fed (Fig. 5) mice, demonstrating the detections of physiological β-cell proliferation. In addition, ratios of Ki67-positive β-cells and plasma Gluc activities showed strong correlations under all of the in vivo experimental conditions employed in this study. Therefore, β-cell proliferation with different characteristics are well detected with the Ki67p-Gluc system both quantitatively and temporally. These results indicate that the in vivo Cre-induced Ki67p-Gluc system established in this study is a highly sensitive method for monitoring cell proliferation in vivo.

Herein, we used the Cre-loxP system to monitor the proliferation of targeted cells. Indeed, by crossing Ki67p-LSL-Gluc mice with different promoter-driven Cre expressing mouse lines, we were able to monitor the proliferation of two cell types, hepatocytes and pancreatic β-cells. Considering especially the success in achieving in vivo monitoring of physiological β-cell proliferation, given how few of these cells are present at the whole-body level, this system is potentially applicable to monitoring the proliferation of a wide variety of cell types in vivo, even those present in very small numbers.

Using this strategy, we have shown detailed time courses of β-cell proliferation during the juvenile period in the same individuals. In addition, we have obtained the evidence of the physiological differences in β-cell proliferation between light and dark periods during the juvenile period in vivo. β-cell proliferation was maintained at high levels during the light period and decreased prior to the beginning of the dark period. Meanwhile, β-cell proliferation reached the nadir during the dark period and gradually increased toward the start of the light period (Fig. 6B). Synthesizing a vast amount of insulin protein and appropriately secreting it in response to stimulations, such as food intake, to maintain whole-body glucose homeostasis are highly characteristic of pancreatic β-cells. Therefore, β-cells consume a large amount of energy to perform these functions. In this respect, during the dark period when the animals actively feed and require insulin secretion, sparing the energy which is expended for cell proliferation may be favorable for efficient production and secretion of insulin. Interestingly, it has been reported that, during β-cell regeneration after extreme β-cell depletion by diphtheria toxin A-mediated ablation, preferential β-cell proliferation was observed during the dark period[49], unlike β-cell proliferation during the juvenile growth stage shown in the present study. These results suggest that diurnal control of β-cell proliferation differs between growth and regenerative processes.

In addition, employing this system, we demonstrated the feasibility of screening for drugs designed to increase cell mass. Gluc is stable in culture media in which its half-life is reportedly around 6 days[1]. The long half-life of Gluc in culture media may allow us to detect cumulative cell proliferation, whenever it occurs within the entire stimulation time, e.g., even if it occurs only for short periods. On the other hand, using gene expression analysis, we can detect the cell proliferation status only at the timepoint when the cells are collected. Therefore, this Gluc ex vivo screening method seems to be optimally sensitive for exploring cell proliferative factors. Indeed, when islets isolated from iβKi67p-Gluc mice were treated with GLP-1 for 96 h, luciferase activities were clearly elevated in the culture media (Fig. 7C), but cell proliferation was not detectable using *MKi67* gene expression analysis at the timepoint of 96 h (Fig. 7D). Thus, this system is applicable to exploring β-cell proliferative factors with the aim of developing strategies for increasing β-cell mass. Since β-cell proliferation capacity is reportedly lower in humans than in mice[42,50], careful consideration is required when applying the results obtained from βKi67p-Gluc mice to human situations. However, several studies have found β-cell mass to be larger in obese than in lean subjects[51–54], indicating low but adequate proliferation capacity in humans as well. Therefore, activating this capacity may lead to a curative therapy for diabetes. At present, the islet cell supply from human donors is insufficient and individual differences among human islet cells remain a challenge when screening for β-cell proliferative factors. In this regard, screening β-cell proliferative factors using islet cells isolated from our β-cell-specific Ki67p-Gluc mice and administering the factors thereby identified to these mice might hold promise as an efficient strategy. In addition to pancreatic β-cells, by crossing Ki67p-Gluc mice with various Cre mice, this strategy enables us to monitor the proliferative status of various types of cells and is applicable to screening trophic factors for these various cell types, e.g., β-cells for diabetes, hepatocytes for liver dysfunction, endocrine cells for various deficiencies, cardiomyocytes for cardiac failure, myocytes for sarcopenia, vascular cells for circulation insufficiencies, and so on. In addition, when applied to cancer cells, this system may be useful when searching for factors suppressing cellular proliferation.

In conclusion, we have developed a highly sensitive strategy which enables real-time monitoring of the proliferation of targeted cell types in vivo, in individual living mice using a simple blood analysis approach. Furthermore, this system allows us to detect cell proliferation ex vivo in a very simple and highly sensitive manner. These strategies may contribute to major advancements in many areas of biological and medical research.

## Methods
### Ethical statement
All experiments in this study were conducted in accordance with the Tohoku University institutional guidelines. Ethics approval was obtained from the Institutional Animal Care and Use Committee of the Tohoku University Environmental & Safety Committee.

### Gene targeting in ES cells and generation of knock-in mice
pGLuc Basic-1 vector was purchased from NanoLight Technologies (Pinetop, AZ, USA). It carries a promoterless reporter gene, Gluc (the humanized coding sequence for the secreted *Gaussia princeps* luciferase)[3]. Proximal promoter of the human Ki67 gene (Ki67p) was PCR amplified using human genome (Agilent Technologies, Santa Clara, CA, USA) as reported previously[13]. To develop a construct for monitoring real-time cell proliferation, the human Ki67 promoter sequence and the loxP-chloramphenicol acetyltransferase (CAT)-polyA-loxP (LSL) cassette were inserted between the *EcoRI* and *HindIII* sites and at the *SacI* site, respectively, upstream from Gluc reporter gene of the pGLuc Basic-1 vector. Ki67p-LSL-Gluc construct-containing vectors were transfected into a mouse hepatocyte cell line, Hepa1-6, or a mouse β-cell line, MIN6. After confirming that the construct worked in these cells, the Ki67p-LSL-Gluc was excised and knocked into the Rosa26 locus in ES cells from C57BL/6 N mice by electroporation. Homologous recombinant ES cell clones were identified by PCR and Southern blot analysis with a neo probe. A chimeric mouse was produced from these established homologous recombinant ES cells. The chimeric mice were crossed with wild type mice and heterozygous mice were thereby obtained (Trans Genic Inc., Hyogo, Japan).

## Animals

The mice were maintained on a 12 h:12 h light:dark cycle at 25 °C with 50% humidity. Lights on is defined as Zeitgeber time (ZT) 0 and lights off as ZT 12. The mice were fed ad libitum a standard laboratory chow diet (MF, 65% carbohydrate, 4% dairy fat, 24% protein; provided by Oriental Yeast, Tokyo, Japan).

The albumin-Cre-ER transgenic mice[25] were kindly provided by Prof. Pierre Chambon and Prof. Daniel Metzger (Institute of Genetics and Molecular and Cellular Biology, Illkirch-Cedex, France). The rat insulin 2 promoter-Cre-ER (RIP-Cre-ER) mice[14] and the rat insulin 2 promoter-Cre (RIP-Cre) mice[55] were purchased from The Jackson Laboratory (Bar Harbor, ME, USA).

To obtain tamoxifen-inducible liver-specific Ki67p-Gluc mice (iLKi67p-Gluc mice), we crossed albumin-Cre-ER mice and Ki67p-LSL-Gluc mice. At 8 weeks of age, the male albumin-Cre-ER; Ki67p-LSL-Gluc mice were injected intraperitoneally with tamoxifen (Sigma, St. Louis, MO, USA), at 80 µg/g body weight, dissolved in corn oil (Sigma) every 24 h for 5 consecutive days. At 10 days after the last injections, 70% partial hepatectomy (PHx) was performed on these mice, in accordance with a published protocol[56]. The protocol was for ligation and removal of the left and median lobes of the liver. The sham operation consisted of a midline laparotomy alone.

To obtain tamoxifen-inducible β-cell-specific Ki67p-Gluc mice (iβKi67p-Gluc mice), we crossed RIP-Cre-ER mice and Ki67p-LSL-Gluc mice. At age 8 weeks, the RIP-Cre-ER; Ki67p-LSL-Gluc mice were injected intraperitoneally with tamoxifen (Sigma) as described above. For the hepatic ERK activation model, male iβKi67p-Gluc mice were intravenously injected with $1 \times 10^8$ plaque-forming units (PFU) per mouse of MEK adenovirus or $1 \times 10^8$ PFU per mouse of LacZ adenovirus one month after the last tamoxifen injection. For the pregnancy model, one month after the last injection of tamoxifen, female iβKi67p-Gluc mice were crossed with male C57BL/6 N mice. Male C57BL/6 N mice were purchased from Japan SLC (Shizuoka, Japan). We confirmed mating by the presence of a vaginal plug in the morning after the mice had been left overnight in the same cage. For high fat diet loading, male iβKi67p-Gluc mice were fed high fat chow diet (D12492, 24% protein, 20% carbohydrate and 60% fat; provided by Research Diets) for two months, beginning at 13–14 weeks of age. For glucose tolerance tests, mice were fasted for 10 hours and intraperitoneally injected with 2 g/kg glucose, followed by measurement of blood glucose[57]. Plasma insulin levels were measured using a Mouse Insulin ELISA Kit (Morinaga, Tokyo, Japan).

To obtain congenital β-cell-specific Ki67p-Gluc mice (cβKi67p-Gluc mice), we crossed RIP-Cre mice and Ki67p-LSL-Gluc mice.

## Recombinant adenovirus

Recombinant adenoviruses encoding the Cre recombinase gene under the control of the CAG promoter (Cre-adenovirus) were used. Recombinant adenoviruses were propagated in HEK293 cells and purified by cesium chloride gradient ultracentrifugation. Titers of virus stock were determined by the end point cytopathic effect assay[58,59]. Cells were incubated for 48 hours in media containing $2.2 \times 10^5$ PFU/well of Cre- and LacZ-adenovirus

For hepatic ERK activation, recombinant adenovirus encoding the constitutively active mutant of the Xenopus MEK1 gene (MEK-adenovirus) was used. Similarly to Cre-adenovirus, MEK-adenovirus were propagated, purified, and determined their titers[20,21]. Male iβKi67p-Gluc mice were intravenously injected with $1 \times 10^8$ plaque-forming units (PFU) per mouse of MEK adenovirus. Recombinant adenoviruses encoding the LacZ gene (LacZ-adenovirus) were used as control.

## In vitro studies

Hepa1-6 cells were purchased from ATCC (#CRL-1830) and were maintained in Dulbecco's modified Eagle medium (Thermo Fisher Scientific, Waltham, MA, USA) supplemented with 10% fetal bovine serum, plus penicillin (100 U/mL) and streptomycin (100 mg/mL). MIN6 cells were provided from Prof. Junichi Miyazaki (Osaka University) and were maintained in Dulbecco's Modified Eagle Media containing 25 mM glucose supplemented with 10% fetal bovine serum, plus penicillin (100 U/mL) and streptomycin (100 mg/mL). For adenovirus infection, cells were incubated for 48 hours in media containing $2.2 \times 10^5$ PFU/well of Cre- and LacZ-adenovirus. After incubation, we replaced culture media with new media not containing these viruses and incubated cells for one more hour, and then collected the cells and media. To suppress cell proliferation, Hepa1-6 cells were incubated for 8, 24, 48, 72 and 96 h in media containing mitomycin C (30 µg/mL), roscovitine (500 µg/mL), or rapamycin (50 µM), and then for an hour in media not containing these agents. Each solvent was used as the vehicle.

## Gaussia luciferase assay

To assay cell lysates, the cells were harvested and treated with a surfactant[3]. Blood samples were drawn by making a small incision in the tails of awake mice at ZT 8. A total of 1 µL of 0.5 M EDTA was added to 50 µL of blood, followed by centrifugation for more than 10 min to separate the plasma. Then, 100 µL of 10 µg/mL coelenterazine (Nanolight Technologies) diluted in phosphate buffered saline (PBS) were added to 10- or 20-µL of the samples (culture media, cell lysates, or plasma) in a 96-well plate. Gluc activity was measured using a plate luminometer (Fluoroscan Ascent® FL, Thermo Fisher Scientific) which was set to inject the coelenterazine and to acquire photon counts for 10 s[1].

## Evaluation of gene expression levels by quantitative RT-PCR

Total RNA was extracted using an RNeasy Micro Kit (QIAGEN, Tokyo, Japan) from islets or cell lysates. cDNA synthesized from 100 ng of total RNA with a QuantiTect Reverse Transcription Kit (QIAGEN) was evaluated with a real-time PCR quantitative system (Light Cycler software; Roche Diagnostics, Mannheim, Germany)[60]. Relative amounts of mRNA were calculated with Actb mRNA as the invariant control. The sequences of primers used are listed in Supplementary Table 1.

## In vivo bioluminescence imaging

Mice were anesthetized and injected intravenously with coelenterazine diluted in PBS at a dose of 4 mg/kg. Gluc imaging was performed 1 minute after injection and photon counts were recorded during a 1-min period using a cooled CCD camera (IVIS SPECTRUM, PerkinElmer, Waltham, MA, USA) with no illumination[1]. We manually defined Regions of Interest (ROI) and then calculated the sum of the photon counts in these ROI.

## Islet studies

Pancreatic islet isolation was performed one month after the last injection. Islets were isolated by retrograde infusion of 1.0 mL of cold Hanks' balanced salt solution containing 1.0 mg/mL collagenase V (Sigma) into the pancreatic duct. Pancreases were digested in a thermostat chamber at 37 °C. Purification of the islets from mice was achieved by hand-picking under a light microscopic view. Isolated islets were maintained overnight at 37 °C with 5% $CO_2$ and 95% air in RPMI1640 media containing 10% fetal calf serum, 25 mM glucose, 100 U/mL penicillin, 100 µg/mL streptomycin, and 50 µg/mL gentamicin[57]. The next day, isolated islets were treated with 1.0 mg/mL collagenase for 10 min at 37 °C to assure that peptides reached the inner portions of the islets[21], followed by washing twice with PBS. Then, islets were incubated for 48 h, 72 h, or 96 h with mitogenic stimuli. For stimulation with a high glucose concentration, islets were incubated in RPMI1640 media containing 15- or 25-mM glucose continuously. RPMI1640 media containing 5.5 mM glucose were used for the low glucose concentration experiments. For stimulation with GLP-1, islets were incubated in RPMI1640 media containing 25 mM glucose and

1 mM Diprotin A (Peptide Institute, Osaka, Japan), a dipeptidyl peptidase-4 inhibitor, with 100 nM GLP-1 (Aviva Systems Biology, San Diego, CA, USA). Water was used as the vehicle.

For insulin secretion studies, isolated islets were maintained overnight in RPMI1640 medium containing 10% fetal calf serum, 25 mM glucose, 100 U/mL penicillin, 100 μg/mL streptomycin, and 50 μg/mL gentamycin at 37 °C with 5% $CO_2$ and 95% air. The following day, batches of 10 islets were pre-incubated at 37 °C for 30 min with Krebs-Ringer bicarbonate HEPES buffer (KRBH; 135 mM NaCl, 3.6 mM KCl, 0.5 mM $NaH_2PO_4$, 0.5 mM $MgCl_2$, 1.5 mM $CaCl_2$, 2 mM $NaHCO_3$, 10 mM HEPES and 0.1% BSA) containing 1.67 mM glucose, then incubated for 60 minutes in KRBH with 1.67- or 16.7-mM glucose, or 1.67 mM glucose plus 30 mM KCl. Insulin contents were measured after acid-ethanol treatment (77.5 mM HCl in 75% ethanol) extraction.

## Immunoblotting

Hepa1-6 samples were boiled in Laemmli buffer containing 10 mM dithiothreitol at 100 °C for 5 min, subjected to sodium dodecyl sulfate-polyacrylamide gel electrophoresis and transferred onto nitrocellulose membranes, then blocked in Tris-buffered saline with 3% fetal bovine serum. Nitrocellulose membranes were incubated with rabbit primary antibodies to Gaussia luciferase (401 P, NanoLight Technologies), at a 1:1,000 dilution and then incubated with a secondary horseradish peroxidase (HRP)-conjugated donkey anti-rabbit antibodies (NA9340, GE Healthcare, Tokyo, Japan) at a 1:1,000 dilution. These antibodies were dissolved in Can Get Signal (Toyobo).

## Immunohistochemistry

Hepa1-6 or MIN6 cells cultured in 8-well chamber slides were blocked with 4% PFA for 20 min. For Ki67- or PHH3-positive cell in situ detection, the specimens were stained with rabbit primary antibodies to mouse Ki67 (ab15580; Abcam, Cambridge, UK) at a 1:1000 dilution or rabbit primary antibodies to mouse PHH3 (#9701; Cell Signaling Technology, Danvers, MA, USA) at a 1:200 dilution. Alexa Fluor 594 goat anti-rabbit antibody (111-585-144, Jackson ImmunoResearch Laboratories, West Grove, PA, USA) at a 1:500 dilution or Alexa Fluor 594 goat anti-rabbit antibodies (#8889, Cell Signaling Technology) at a 1:500 dilution were used as secondary antibodies. The number of Ki67 or PHH3-positive nuclei per 500 cells per chamber was counted.

Pancreatic or hepatic tissues were excised and fixed in 10% formalin at 4 °C overnight and embedded in paraffin. Pancreatic or liver sections 3 μm in thickness were made at an interval of 250 μm.

For Ki67- or PHH3-positive hepatocyte in situ detection, the specimens were stained with rabbit primary antibodies to mouse Ki67 (#12202, lot 6; Cell Signaling Technology) at a 1:400 dilution or rabbit primary antibodies to mouse PHH3 (#9701; Cell Signaling Technology) at a 1:200 dilution. HRP-labeled goat anti-rabbit antibodies (#8114; Cell Signaling Technology) at a 1:500 dilution was used as the secondary antibody. The immune complexes were visualized with 3,3′-Diaminobendizine (DAB, #040-27001, WAKO, Osaka, Japan) solution. The number of Ki67- or PHH3-positive nuclei per 3000 hepatocytes per liver specimen was counted.

For Ki67- or PHH3-positive pancreatic β-cell in situ detection, the specimens were stained with rabbit primary antibodies to mouse Ki67 (ab15580, Abcam) at a 1:100 dilution, rabbit primary antibodies to mouse PHH3 (#9701; Cell Signaling Technology) at a 1:200 dilution and guinea pig primary antibodies to mouse insulin (IR002, Agilent Technologies) without being diluted, respectively. Alexa Fluor 594 conjugate goat anti-Rabbit antibodies (711-585-144, Jackson ImmunoResearch Laboratories) at a 1:200 dilution and Alexa Fluor 488 goat anti-guinea pig antibodies (ab150117, Abcam) at a 1:200 dilution were used as secondary antibodies. The number of Ki67- or PHH3-positive nuclei per 3000 insulin-positive cells per pancreas specimen was counted.

## Statistics and reproducibility

All data were expressed as means ± SEM. The statistical significance of differences between two groups was assessed employing the unpaired t test. The statistical significance of differences between two variables for the same subject was assessed employing the paired t test. For experiments involving three or more groups, one-way analysis of variance (ANOVA) was used followed by Bonferroni's post hoc test. Statistical analyses of the luciferase activity time courses were performed by repeated-measures one-way ANOVA followed by the Tukey multiple comparison test. Pearson's correlation coefficient was used to determine the correlation between quantitative two variables. Statistical analyses were performed using Microsoft Excel. No statistical method was used to predetermine the sample size. No data were excluded from the analyses. The experiments were not randomized. The Investigators were not blinded to allocation during experiments and outcome assessment.

## Reporting summary

Further information on research design is available in the Nature Portfolio Reporting Summary linked to this article.

## Data availability

The data supporting the findings in this paper can be found in the main Article file, Supplementary Information, and Source Data file. Source data are provided with this paper.

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

## Acknowledgements

Profs. Pierre Chambon and Daniel Metzger of the Institute of Genetics and Molecular and Cellular Biology, Illkirch-Cedex contributed to generation of the SA-Cre-ER$^{T2}$ mice.

This work was supported by Grants-in-Aid for Scientific Research to J.I. (22H03124 and 22K19303) and H.K. (20H05694) from the Japan Society for the Promotion of Science. This research was also supported by the Japan Agency for Medical Research and Development, AMED, under Grant Numbers 21gm6210002h0004 (AMED-PRIME) (to J.I.) and JP20gm5010002h0004 (to H.K.), as well as by the Japan Science and Technology Agency, JST [Moonshot R&D] [Grant Number JPMJPS2023] (to H.K.).

We thank Ms. T. Takasugi, K. Watanabe, S. Goto, Y. Yoshizawa, K. Takahashi M. Iwama and H. Yokoyama (All belong to the Department of Metabolism and Diabetes, Tohoku University Graduate School of Medicine) for technical support.

## Author contributions

H.S., J.I., and J.Y. conducted the research and obtained the data, contributed to relevant discussions, wrote/reviewed/edited the manuscript. T.I., Y.K., A.E., M.K., J.S., H.Kubo, H.Komamura, Y.M., Y.A., S.H., S.S., S.K., K.T., and K.K. contributed to the relevant discussions. H.Katagiri. contributed to the relevant discussion, wrote/reviewed/edited the manuscript.

## Competing interests

The authors declare no competing interests.
