## [Peer Review File · Nature Communications]

Reviewers' Comments:

Reviewer #1:

Remarks to the Author:

The authors of this paper claim to have developed a novel tool to monitor cell proliferation, both in vitro and in vivo, in the latter case, in mouse models in a tissue-specific manner and with little invasion. Whereas such a tool might be useful in specific circumstances, its establishment would require a rigorous assessment of cell proliferation in parallel, which unfortunately is missing throughout all the experimental settings; thus, the authors have not validated that the tool functions as claimed. Moreover, the authors claim that, thanks to this tool, important results were generated (circadian rhythm of beta-cell proliferation), but as these have also not been controlled, they remain speculative. Thus the paper remains largely methodological and descriptive.

The authors introduced the proximal promoter of human Ki-67 (Ki67p) upstream of humanised secreted Gaussia luciferase (GLuc). Floxed Poly-A cassette upstream of the GLuc reporter acts as a stop sequence in the absence of Cre recombinase. Cre is introduced either by infecting cells transfected with Ki67p-GLuc with Cre-adenovirus (in vitro experiments), or by crossing Ki67p-GLuc mice (transgene in the Rosa26 locus) with tissue-specific Cre-expressing mice (in vivo experiments). Ki-67 is a widely-used cell proliferation marker, whose expression is under the control of cell cycle machinery, thus mimicking cyclin expression (there exist more appropriate references than the cited ref. 18 that studied the regulation of Ki-67 expression), and which is absent in terminally differentiated quiescent or senescent cells. The proximal Ki-67 promoter contains E2F transcription factor binding site, and the study of Zamboni (ref. 19) showed that the expression of a downstream reporter gene can thus be cell cycle-regulated. In the study of Sugawara et al., cells produce Gaussia luciferase, which is secreted, and thus its concentration can be quantitatively measured either in cell culture media, or in mouse blood samples. However, equating this with cell proliferation rates assumes that the entire control of the levels of the GLuc is due to cell cycle-related transcription with no influence of controls of translation, degradation, secretion, or anything else; and that the luciferase activity measured linearly correlates with proliferation. Nowhere is this demonstrated. Yet the authors use various in vitro and in vivo systems in which they measure GLuc activity from which they infer cell proliferation status.

Major problems:

1. In none of the presented experiments is the GLuc concentration controlled with cell proliferation.

a. For example, in Fig. 1, in the two cell lines analysed, the GLuc concentration increases upon Cre induction, but this does not mean that it necessarily reflects cell proliferation. Astonishingly, the authors instead state: "Cell lines are expected to actively proliferate." Similarly, cell proliferation inhibitors, such as mitomycin C, rapamycin and roscovitine are used without controlling for their effect on cell proliferation. Moreover, between the two cell lines in Fig. 1B and C, and between the different treatments of the same cell line in D-F, the relative GLuc activity varies by a factor of 10 and more, but why this is the case and to what this could be due is not addressed nor explained. Instead, the authors summarise that « These findings indicate that GLuc activity in the culture media reflects the cell proliferation status of these cells »

b. Similarly, in in vivo experiments (Fig. 2 and 3), GLuc measurements in blood samples of mice were in no way correlated with cell proliferation in order to provide evidence that the GLuc levels reflect cell proliferation in a quantitative manner. For example, partial hepatectomy (PH) experiments (Fig. 2) were performed in order to stimulate hepatocyte proliferation, in mice expressing the Cre under the albumin promoter, specific for hepatocytes. Then, the authors cite a 1997 paper, in which kinetics of liver cell proliferation after PH was performed, and use these results to validate their measurements of GLuc in mice blood. There are several problems with such an approach. Even if the objective of the study is to provide a non-invasive tool for in vivo monitoring of cell proliferation, to be valid, such a tool needs to be first confirmed by analysing cell proliferation (by the most conventional tools, like immunohistology analysis of Ki-67 staining), even if this would require sacrificing mice, and in experiments performed by the same lab, and not by citing 20 year-old studies. Secondly, in the cited paper (ref. 20), hepatocyte proliferation occurs within the first 48hrs after PH, and thus it does not quite correspond to the results obtained with GLuc measurements. Indeed, these are other liver cells that continue to proliferate till day 5, but

they do not express albumin. There is no discussion of, or explanation for, these discrepancies. c. Similarly, in experiments meant to follow pancreatic beta-cell proliferation in mice, there is no control for the actual cell proliferation, and either previous publications are cited to serve as controls (for MEK pathway activation, Fig. 3A, B, or pregnancy, Fig. 3C, D), or the elevated level of GLuc was equated with increased cell proliferation without any evidence provided (high fat diet loading, Fig. 3E, F). In contrast, mRNA levels of Ki-67 were measured in isolated islet cells in the experiments meant to evaluate beta-cell proliferation in juvenile mice (Fig. 4A and Suppl. Fig. 4A). If the mice were sacrificed in those experiments, why staining for Ki-67 was not performed to actually count proliferating cells, and why such experiments were not done as a control in other conditions, is unclear.

2. The conclusion drawn from the experiment involving measurement of GLuc levels every 4-hours seems to imply that either beta-cells divide very rapidly and stop dividing at night, which would correspond to decreased GLuc plasma levels, or that all cells are synchronised and are in their early G1 phase at night, which would correspond to low Ki-67 expression (and thus low GLuc expression). Therefore, inferring that fluctuations in GLuc levels during the day correspond to cell proliferation changes due to the circadian rhythm is an overinterpretation.

3. The reasoning behind the last set of experiments is hard to understand. There exists a plethora of simple methods to follow cell proliferation in cell cultures and so why one would need to measure GLuc concentration in media to deduce whether cells proliferate or not, is hard to justify, especially given that GLuc persists in media even if cells stop proliferating (even assuming that measuring mRNA levels of Ki-67 is the right way to assess cell proliferation, which it is not).

Minor points:

1. The quality of figures is very low from the aesthetic point of view.
2. It would be easier for the reader if the graphs, for example in Fig. 1, were labelled with cell type names.
3. It is not clear why data of Fig. 1A, B and C-E are analysed with two different statistical tests.
4. Fig. 2B, no quantification of luciferase bioluminescence has been done.
5. Fig. 3A, the authors state that the luciferase activity remains stable after day 5, while in the graph, a continuous decrease is seen till the end of the experiment.
6. It would require verifying whether data on Fig. 3F (thus E) are indeed more significantly different than in Fig. 3D (thus C).
7. To test whether the insulin secretion does not interfere with Gluc secretion, the authors perform glucose loading test and conclude that there is no hindrance with Gluc secretion (Suppl. Fig. 3). However, as this was conducted in conditions where Gluc is not expected to be expressed (no induction of beta-cell proliferation), this control is meaningless.
8. There is no control for MEK activation (Fig. 3A, B).
9. There are various general statements that are meaningless, like "These novel technologies thus may contribute to advancements in numerous areas of biological research." Such as?

In summary, the paper of Sugawara et al. falls short of the authors' claims. They analyse levels of GLuc in cell culture media and in blood samples of mice without providing any evidence that they in fact correlate with cell proliferation. The number of experiments performed is very limited, most of them lack the required controls, and the results are overinterpreted. Thus, the paper remains essentially descriptive and does not deliver any novel technology nor biologically relevant insights. We unfortunately cannot support the paper for publication.

Reviewer #2:

Remarks to the Author:

In this manuscript, the authors developed an in vivo strategy for quantitative monitoring of the time course of β -cell proliferation status over time in the same system, based on the activity of the secreted *Gussia* luciferase in the blood/plasma of animals. Since pancreatic β -cells constitute only a small population among whole-body cells and have limited proliferative capacity, the authors postulate that this strategy would potentially be applicable to detecting the proliferation status of a variety of cell types. For monitoring of targeted cell proliferation, the authors produced a knock-in mice in which Gluc is expressed under the control of Ki67 promoter (induced during cell

proliferation), Separated by a LoxP, and therefore Gluc is expressed only in proliferative cells expressing Cre. By crossing these mice with Cre recombinase, one can monitor proliferation status of "targeted" cell type in vivo. They used the reporter to monitor liver and β -cell proliferation.

the developed system is very clever, simple and could be widely used.

Authors assayed conditioned medium of cells. It would be important to measure the levels of extracellular versus intracellular Gluc levels at any given time point by assaying its activity and by Western blotting. The authors refer to previous studies that 90% of Gluc is in conditioned media, but this is true for wild-type Gluc and should be tested in their system.

In figure 1, The authors use different treatment to show that the reporter reflects proliferation status of the cells, but only measure activity at one time point. Monitoring at different time points (0,8,24,48,72,96 etc) should be performed with different treatments/controls to confirm that the reporter can monitor cell proliferation over time.

Would help to add the "cell name" or treatment used in each figure panel

Bioluminescence images presented in figure 2B are very naïve. Only a single mouse is shown and at one time point. Imaging should be taking at different time points and results from multiple mice should be presented.

The rationale that insulin secretion might affect Gluc secretion is not clear; many other proteins are secreted from the cells. Why focus on insulin.

In Figure 3, authors monitored Gluc reporter over time. It would be important to correlate the Gluc signal with a typical proliferation assay ex vivo (such as immunostaining and others) to prove that the produced Gluc signal is indeed in response to cell proliferation. this comment goes to all in vivo models used in this study.

For in culture work, both figure 1 and figure 5, cell proliferation should be monitored by different assays and correlated to Gluc signal

Reviewer #3:

Remarks to the Author:

In this manuscript Sugawara et al. report generation of a novel mouse model with secreted type of luciferase (Gluc) engineered to be expressed under the control of Ki67 promoter in tissue/cell type-specific manner (i.e. driven by Cre expression). Generation of this mouse model permits detection of cell specific proliferation events by measurements of blood luciferase concentrations in a relatively small sample volume (~10-20ul). Authors subsequently demonstrate that increases in hepatocyte and beta cell proliferation associated with homeostatic perturbations such as hepatectomy, obesity, pregnancy, etc can be detected and quantified longitudinally in the same mouse by measuring secreted form of luciferase in blood. Authors also report that this methodology also detects diurnal patterns in beta cell proliferation.

General Comments:

This work describes generation of a novel mouse model that has potential to be adapted by beta cell field in the future and provide methodological advantages to existing methods. Considering these advantages, there are 2 general concerns that attenuate my enthusiasms for this manuscript. Firstly, the manuscript doesn't provide any novel information related to the regulation of beta cell proliferation. The study confirms that beta cell proliferation in mice is induced in response to obesity, pregnancy, etc, but this information has been known for decades. Authors suggest that circadian regulation of beta cell proliferation is a novel observation, however this has been reported by others using standard techniques and much more rigorous mechanistic approaches (e.g. Genes Dev 34, 1650-1665 (2020)). Secondly, the rate of beta cell proliferation in

human beta cells is characteristically low and in contrast to mouse models, human beta cells do not appear to augment proliferative rate in response to common metabolic perturbations (e.g. obesity and pregnancy). Thus, the use of mouse models to study mechanisms regulating beta cell proliferation in obesity and diabetes has long been questioned.

Specific Comments:

1. Increase in luciferase activity (as an indicator of beta cell proliferation) has to be validated by objective measures of beta cell proliferation (e.g. protein expression of Ki67, PCNA or BrdU by immunodetection methods). The existing immunodetection methods (e.g. Ki67 staining by immunohistochemistry) allows for quantification of the frequency or the rate of beta cell proliferation (e.g. % proliferating beta cells). Correspondingly, existing methodology has shown a several fold increase in the frequency of beta cell proliferation in response to obesity/pregnancy. It will be important to compare/contrast how an increase in the frequency of beta cell proliferation compares to fold induction of luciferase activity presented in the current study.
2. Beta cell secretory pathway is altered in response to conditions associated with obesity and Type 2 diabetes. This is related to commonly associated abnormalities in beta cell ER function, protein folding, autophagy, etc. Thus, authors should confirm that expected changes in ER and secretory functions in "stressed/diabetic" beta cells will not interfere with Gluc production/secretion and thus result in inaccurate approximation of beta cell proliferation using this method.
3. Authors should also provide data on glucose and KCL-stimulated insulin secretion as well as insulin content in isolated islets of $i\beta$ Ki67pGluc mice. This data should be presented and compared to proper controls (e.g. RIP-Cre and floxed mice) and is important to make sure there are no alterations in beta cell secretory function in this model.

Reviewer #4:

None

Reviewer #5:

None

Responses to Reviewer 1

there exist more appropriate references than the cited ref. 18 that studied the regulation of Ki-67 expression

As suggested, we have added additional references regarding the regulation of Ki-67 expression (Gerdes et al, 1984, *J Immunol*, Scholzen et al, 2000, *J. Cell. Physiol*) as Refs 10 and 11 to the revised manuscript.

Major problems:

1. In none of the presented experiments is the GLuc concentration controlled with cell proliferation.

a. For example, in Fig. 1, in the two cell lines analysed, the GLuc concentration increases upon Cre induction, but this does not mean that it necessarily reflects cell proliferation. Astonishingly, the authors instead state: "Cell lines are expected to actively proliferate." Similarly, cell proliferation inhibitors, such as mitomycin C, rapamycin and roscovitine are used without controlling for their effect on cell proliferation. Moreover, between the two cell lines in Fig. 1B and C, and between the different treatments of the same cell line in D-F, the relative GLuc activity varies by a factor of 10 and more, but why this is the case and to what this could be due is not addressed nor explained. Instead, the authors summarise that « These findings indicate that GLuc activity in the culture media reflects the cell proliferation status of these cells

We appreciate the reviewer's important comments on the necessity of controls for cell proliferation status. Therefore, to determine whether Gluc activity in the culture media actually reflects the cell proliferation status, we analyzed *MKi67* gene expressions of Hepa1-6 cells which had been treated with cell cycle inhibitors along with measuring medium Gluc activities. First, *MKi67* gene expressions were not altered by Cre-adenovirus in Hepa1-6 cells, indicating that Cre expression does not affect the basal cell proliferation status of these cells. On the other hand, three types of cell cycle inhibitors, Mitomycin C, Rapamycin and Roscovitine markedly decreased expressions of *MKi67* consistently with Gluc activities in the culture media. In addition, when we compared *MKi67* gene expressions of cells and Gluc activities in culture media in these experiments, these two parameters showed a marked and significant correlation ($r = 0.903$, $P < 0.0001$). Thus, Gluc activities in the culture media do, indeed, reflect the cell proliferation status of Hepa1-6 cells.

In the original manuscript, we evaluated Gluc activities in the media after a 20-hour cell culture. In the revised experiment, on the other hand, to evaluate the real time proliferative status of cultured cells, we conducted multiple measurements of Gluc activities in the culture media 1 hour after replacing it. We found that Cre-adenovirus infection increased Gluc activities in the culture media of Hepa1-6 cells to similar extents in all experiments. In contrast, Gluc activities were still higher, in the culture media of Hepa1-6 than in those of MIN6 cells. We compared *MKi67* gene expressions of Hepa1-6 and MIN6 cells, and found that *MKi67* gene expressions of Hepa1-6 cells were similarly higher than those of MIN6 cells. Therefore, high Gluc activities in the culture media of Hepa1-6 cells are due to higher cell proliferation activity in Hepa1-6 than in MIN6 cells. These results further support the notion that Gluc activities in culture media reflect actual cell proliferation status. These findings are now presented in Supplemental Figures 1B and 2A to 2G and are described in the Results section of the revised manuscript (page 5, lines 21 to 25 and page 5, lines 29 to 36).

b. Similarly, in in vivo experiments (Fig. 2 and 3), GLuc measurements in blood samples of mice were in no way correlated with cell proliferation in order to provide evidence that the GLuc levels reflect cell proliferation in a quantitative manner. For example, partial hepatectomy (PH) experiments (Fig. 2) were performed in order to stimulate hepatocyte proliferation, in mice expressing the Cre under the albumin promoter, specific for hepatocytes. Then, the authors cite a 1997 paper, in which kinetics of liver cell proliferation after PH was performed, and use these results to validate their measurements of GLuc in mice blood. There are several problems with such an approach. Even if the objective of the study is to provide a non-invasive tool for in vivo monitoring of cell proliferation, to be valid, such a tool needs to be first confirmed by analysing cell proliferation (by the most conventional tools, like immunohistology analysis of Ki-67 staining), even if this would require sacrificing mice, and in experiments performed by the same lab, and not by citing 20 year-old studies. Secondly, in the cited paper (ref. 20), hepatocyte proliferation occurs within the first 48hrs after PH, and thus it does not quite correspond to the results obtained with GLuc measurements. Indeed, these are other liver cells that continue to proliferate till day 5, but they do not express albumin. There is no discussion of, or explanation for, these discrepancies.

We appreciate with the reviewer's important comments. As suggested, we examined whether plasma Gluc activities actually reflect the hepatocyte proliferative status *in vivo*. We histologically analyzed proportions of Ki67-positive hepatocytes along with

evaluating plasma Gluc activities on days 0, 2 and 9 after partial hepatectomy (PHx) using liver-specific Ki67p-Gluc mice (iLK67p-Gluc mice). Since hepatocytes can be distinguished from other cell types according to the sizes and shapes of their nuclei, we showed the histological images together with Ki67 staining in each experiment. Plasma Gluc activities were markedly increased on day 2, followed by reversal to the basal level on day 9 after PHx. Consistent with these results, proportions of Ki67-positive cells to all hepatocytes were markedly increased on day 2 followed by reversal to the basal level on day 9 after PHx. In addition, when we compared proportions of Ki67-positive hepatocytes in the liver and plasma Gluc activities in these experiments, these two parameters showed a marked and significant correlation ($r = 0.904$, $P < 0.0001$). These results indicate that plasma Gluc activities in iLK67p-Gluc mice quantitatively reflect the hepatocyte proliferative status after PHx *in vivo*. These findings are now presented in Supplemental Figures 2B and C, and are described in the Results section of the revised manuscript (page 6, lines 26 to 31).

As to the second point raised by the reviewer, we agree with his/her comments. Ref 20 of the original manuscript, a review article, cited a very old report which examined hepatocyte proliferation after PHx in rats but not in mice (Grisham et al, *Cancer Res*, 1962). In addition, another previous report showed that hepatocyte proliferation occurred at an earlier timepoint after PHx in rats than in mice (Weglarz et al, *PNAS*, 2000). We previously showed drastically increased hepatocyte proliferation on day 2, with mild hepatocyte proliferation persisting thereafter until day 7 after PHx, by evaluating BrdU-positive hepatocytes in our previous (Ref 23 of the original manuscript; now Ref 24 in the revised manuscript). We have now further confirmed that hepatocyte proliferation persists on day 5 after PHx by staining with Ki67, another cell proliferation marker (Figure A shown below). These findings indicate that the difference in hepatocyte proliferation time courses between the previous report using rats and the previous and present studies using mice is due to a species difference. Again, the aforementioned time course of Ki67 data obtained from hepatocytes revealed plasma Gluc activities in iLK67p-Gluc mice to quantitatively reflect the hepatocyte proliferative status after PHx in mice *in vivo*. Therefore, we removed the inappropriate reference (Ref 20 in the original manuscript) from the revised manuscript. Since the species difference between rats and mice is not the main point of this study, we did not add these results to the revised manuscript. We appreciate the reviewer's comments.

Figure A

c. Similarly, in experiments meant to follow pancreatic beta-cell proliferation in mice, there is no control for the actual cell proliferation, and either previous publications are cited to serve as controls (for MEK pathway activation, Fig. 3A, B, or pregnancy, Fig. 3C, D), or the elevated level of GLuc was equated with increased cell proliferation without any evidence provided (high fat diet loading, Fig. 3E, F). In contrast, mRNA levels of Ki-67 were measured in isolated islet cells in the experiments meant to evaluate beta-cell proliferation in juvenile mice (Fig. 4A and Suppl. Fig. 4A). If the mice were sacrificed in those experiments, why staining for Ki-67 was not performed to actually count proliferating cells, and why such experiments were not done as a control in other conditions, is unclear.

In the experiments conducted to evaluate β -cell proliferation *in vivo* as well, we first examined Ki67-positive β -cells histologically along with evaluating plasma Gluc activities on day 0, 2 and 10 after adenoviral administration in β -cell-specific Ki67p-Gluc mice (i β Ki67p-Gluc mice) treated with adenovirus containing the active mutant gene of MEK-1 (L-MEK mice). Both proportions of Ki67-positive β -cells and plasma Gluc activities were markedly increased on day 2, followed by reversal to the basal level on day 10 after adenoviral administration. In addition, proportions of Ki67-positive β -cells and plasma Gluc activities showed a marked and significant correlation ($r = 0.835$, $P < 0.001$).

We next performed similar experiments in a pregnant model. Plasma Gluc activities and Ki67-positive β -cells were both significantly increased on day 12, followed by significant decrements on day 21, of pregnancy. In addition, proportions of Ki67-positive β -cells and plasma Gluc activities showed a significant correlation ($r = 0.812$, $P < 0.005$).

In addition, we performed similar sets of experiments in a high fat diet (HFD)-induced obesity model. In this model, increases in plasma Gluc activities were mild, and there was a significant difference between the timepoints of days 0 and 8 weeks after starting HFD

in the original manuscript. Therefore, we measured Gluc activities and counted Ki67-positive β -cells at these two timepoints and obtained data indicating that Gluc activities and Ki67-positive β -cells were both significantly increased 8 weeks after starting the HFD. Again, proportions of Ki67-positive β -cells and plasma Gluc activities showed a strong and significant correlation ($r = 0.914$, $P < 0.005$).

These results show clearly that plasma Gluc activities of i β Ki67p-Gluc mice reflect the β -cell proliferative status in all models employed in the present study.

These findings are now presented in Figures 3C, 3D, 4C, 4D, 5C and 5D, and are described in the Results section of the revised manuscript (page 8, lines 4 to 8, page 9, lines 6 to 11 and page 9, lines 23 to 28).

2. The conclusion drawn from the experiment involving measurement of GLuc levels every 4-hours seems to imply that either beta-cells divide very rapidly and stop dividing at night, which would correspond to decreased GLuc plasma levels, or that all cells are synchronised and are in their early G1 phase at night, which would correspond to low Ki-67 expression (and thus low GLuc expression). Therefore, inferring that fluctuations in GLuc levels during the day correspond to cell proliferation changes due to the circadian rhythm is an overinterpretation.

As the reviewer noted, describing this phenomenon as being due to “circadian rhythm” might well be an overinterpretation. Therefore, we removed the “circadian rhythm” descriptions, and instead described “diurnal variation” or “differences in β -cell proliferation between light and dark periods” in the Abstract, Introduction and Discussion sections (page 2, line 11, page 4, lines 25 to 26, page 10, lines 14 to 15, page 11, line 3, page 11, line 6, page 11, line 16 and page 13, lines 35 to 36).

3. The reasoning behind the last set of experiments is hard to understand. There exists a plethora of simple methods to follow cell proliferation in cell cultures and so why one would need to measure GLuc concentration in media to deduce whether cells proliferate or not, is hard to justify, especially given that GLuc persists in media even if cells stop proliferating (even assuming that measuring mRNA levels of Ki-67 is the right way to assess cell proliferation, which it is not).

As the reviewer noted, Gluc is stable in culture media in which its half-life is reportedly around 6 days. The long half-life of Gluc in culture media may allow us to detect cumulative cell proliferation, whenever it occurs within the entire stimulation time, e.g.,

even if it occurs only for short periods. On the other hand, using gene expression analysis, we can detect the cell proliferation status only at the timepoint at which the cells are collected. Therefore, this Gluc *ex vivo* screening method is far more sensitive for exploring cell proliferative factors. Indeed, as shown in Figures 5E and 5F of the original manuscript (now presented as Figures 7C and 7D in the revised manuscript), when we treated islets isolated from $i\beta$ Ki67p-Gluc with GLP-1 for 96 hours, increases in luciferase activity were clearly detected in the culture media, while cell proliferation was not detectable by *MKi67* gene expression analysis at 96 hour the timepoint. In addition to the high sensitivity, by crossing Ki67p-Gluc mice with various Cre mice, this strategy enables us to monitor the proliferative status of various types of cells and is applicable to screening for trophic factors for these cell types. Therefore, this novel and highly sensitive *ex vivo* screening method is expected to provide with broad utility. To make the rationale easily understandable, we revised the explanation in the Discussion section (page 14, lines 16 to 34 in the revised manuscript).

Minor points:

1. The quality of figures is very low from the aesthetic point of view.

We apologize for presenting figures of low quality. We have replaced the pictures of all figures with images of high quality throughout the manuscript.

2. It would be easier for the reader if the graphs, for example in Fig. 1, were labelled with cell type names.

As suggested, we added the names of cell types in Figures 1B to F, as well as Supplemental Figures 1, 2 and 3.

3. It is not clear why data of Fig. 1A, B and C-E are analysed with two different statistical tests.

In Fig1B and C, for analyzing significance of differences between two groups, we employed the unpaired t test. In Fig D to F, as the significance of differences among four groups was analyzed, we employed one-way analysis of variance (ANOVA), followed by Bonferroni's post hoc tests.

4. Fig. 2B, no quantification of luciferase bioluminescence has been done.

As suggested, we examined time course of liver luminescence using multiple iLKi67p-Gluc mice (n=4-5) before and after PHx on days 2 and 9, and quantitatively analyzed the intensities of luminescence at each timepoint. Luminescence signals were significantly increased at the site of the remnant liver on day 2 as compared with those before the PHx, and had then reverted to the basal levels on day 9 after PHx. These results are consistent with those of plasma Gluc activity and further indicate that luciferase activity elevation in plasma is attributable to selective enhancement of Gluc production in the liver after PHx. These findings are now presented in Figures 2E and are described in the Results section of the revised manuscript (page 7, lines 6 to 9).

5. Fig. 3A, the authors state that the luciferase activity remains stable after day 5, while in the graph, a continuous decrease is seen till the end of the experiment.

We appreciate the reviewer's suggestion. We changed the description suggested by the reviewer to "followed by further gradual decreases until day 14" (page 7, lines 33 to 34).

6. It would require verifying whether data on Fig. 3F (thus E) are indeed more significantly different than in Fig. 3D (thus C).

As pointed out, in the original experiments, we obtained smaller *p* values in the HFD-model than in the pregnant model, likely because we analyzed larger number of mice in experiments on the HFD model (n=7-8 in each group) than in those using the pregnant model (n=4-5 in each group). To examine the significance of plasma Gluc activity in pregnant mice more precisely, we performed further experiments by increasing the number of animals (n=8-9 in each group). As a result, *p* values for the comparison of AUC values of the plasma luciferase activities of pregnant and control iβKi67p-Gluc mice became smaller and were similar to those obtained with the HFD model. Therefore, we replaced Figures 3C and 3D of the original manuscript with new figures including the results of additional experiments, and the findings are now presented in Figures 4A and 4B of the revised manuscript.

7. To test whether the insulin secretion does not interfere with Gluc secretion, the authors perform glucose loading test and conclude that there is no hindrance with Gluc secretion (Suppl. Fig. 3). However, as this was conducted in conditions where Gluc is not expected to be expressed (no induction of beta-cell proliferation), this control is meaningless.

We agree with the reviewer's comments that the influence of insulin secretion on β -cell Gluc secretion needs to be evaluated under conditions wherein β -cell proliferation is enhanced. Therefore, we performed glucose tolerance tests on L-MEK mice on day 2 after adenoviral administration, when β -cell proliferation is highly enhanced, followed by monitoring both the insulin level and luciferase activity in plasma samples collected from these mice. Thirty minutes after the glucose administrations, both blood glucose and plasma insulin levels were significantly increased as compared to the pre-glucose loading levels. In contrast, plasma luciferase activity was not altered after glucose loading. Thus, insulin secretion is unlikely to have an impact on the Gluc secretory process in β -cells even when insulin secretion is enhanced. We replaced the GTT data with those of L-MEK mice on day 2 after adenoviral administration (Supplemental Figures 5A to 5C) and described the findings in the Results section of the revised manuscript (page 8, lines 18 to 26).

8. There is no control for MEK activation (Fig. 3A, B).

We used i β Ki67p-Gluc mice treated with LacZ adenovirus as control in those experiments. This is now clearly described in the revised manuscript (page 7, lines 28 to 29).

9. There are various general statements that are meaningless, like "These novel technologies thus may contribute to advancements in numerous areas of biological research." Such as?

Since our Ki67p-Gluc mouse system reported herein is highly sensitive, allowing quantitative detection of the proliferation of a very small cell population, e.g., pancreatic β -cells, by crossing Ki67p-LSL-Gluc mice with different promoter-driven Cre expressing mouse lines, this system may be applicable to screening for trophic factors designed to increase the masses of a variety of cells used for treatments, such as not only β -cells for diabetes, but also hepatocytes for liver dysfunction, endocrine cells for various deficiencies, cardiomyocytes for cardiac failure, myocytes for sarcopenia, vascular cells for circulation insufficiencies, and so on. In addition, when applied to cancer cells, this system may be useful in searching for factors suppressing cellular proliferation. Thus, these novel strategies may contribute to major advancements in many areas of biological and medical research including those of development, growth, tissue repair, organ regeneration and cancer. To promote understanding of the potential of our approach, we

added these descriptions to the Discussion section (page 14, lines 27 to 34).

Responses to Reviewer 2

Authors assayed conditioned medium of cells. It would be important to measure the levels of extracellular versus intracellular Gluc levels at any giving time point by assaying its activity and by Western blotting. The authors refer to previous studies that 90% of Gluc is in conditioned media, but this is true for wild-type Gluc and should be tested in their system.

As suggested, we measured extracellular versus intracellular Gluc levels, both activities and protein levels, using Hepa1-6 cells with the Ki67p-LSL-Gluc construct after Cre-adenovirus infection. We found that Gluc protein and activities were clearly detected in culture medium, while 1.1 and 1.4% of these Gluc levels, respectively, were detected in Hepa1-6 cell lysates. These findings indicate that the vast majority of Gluc produced in the cells are secreted into the culture media. These findings are now presented in Supplemental Figure 1A and are described in the Results section of the revised manuscript (page 5, lines 15 to 19).

In figure 1, The authors use different treatment to show that the reporter reflects proliferation status of the cells, but only measure activity at one time point. Monitoring at different time points (0,8,24,48,72,96 etc) should be performed with different treatments/controls to confirm that the reporter can monitor cell proliferation over time.

As suggested, we examined the time courses of Gluc activities in culture media of Hepa1-6 cells which had been treated with cell cycle inhibitors until the culture period reached 96 hours.

In the original manuscript, we evaluated Gluc activities of culture media in which Hepa1-6 cells were cultured for 20 hours. In the revised experiment, on the other hand, to evaluate the time courses of Gluc activities in culture media, we replaced the culture media with new media 1 hour before the evaluation timepoints, i.e., at 0, 8, 24, 48, 72 and 96 hours after starting treatments with cell cycle inhibitors. We thereby observed that Gluc activities began to decrease from 8 hours after starting cell cycle inhibitor treatments, and thereafter, almost complete inhibition of these activities persisted from 24 to 96 hours after starting cell cycle inhibitor treatments. These findings were consistent with those of *MKi67* gene expression and these parameters showed a strong and significant correlation

($r = 0.903$, $P < 0.0001$). These results clearly demonstrated that Gluc activities in culture media reflect the cell proliferative status of Hepa 1-6 cells. These findings are now presented in Supplemental Figures 2A to 2G and are described in the Results section of the revised manuscript (page 5, lines 29 to 36).

Would help to add the “cell name” or treatment used in each figure panel

As suggested, we added the names of the cell types used to Figure 1B through F as well as to Supplemental Figures 1, 2 and 3.

Bioluminescence images presented in figure 2B are very naïve. Only a single mouse is shown and at one time point. Imaging should be taking at different time points and results from multiple mice should be presented.

As suggested, we examined the time courses of liver luminescence using multiple iLKi67p-Gluc mice ($n=4-5$) before and after partial hepatectomy (PHx) on days 2 and 9, and then evaluated the intensity of luminescence quantitatively. Luminescence signals were significantly increased at the site of the remnant liver on day 2 as compared with those before the PHx, and had reverted to the basal levels on day 9 after PHx, which is consistent with the results of plasma Gluc activity. These results further indicate that luciferase activity elevation in plasma is attributable to selective enhancement of Gluc production in the liver after PHx. These findings are now presented in Figure 2E and are described in the Results section of the revised manuscript (page 7, lines 6 to 9).

The rationale that insulin secretion might affect Gluc secretion is not clear; many other proteins are secreted from the cells. Why focus on insulin.

Unlike other cell types, pancreatic β -cells are the only type of cells producing insulin. Each β -cell produces and secretes a large amount of insulin to meet whole-body insulin demand. In addition, insulin secretion is further enhanced in situations such as hepatic ERK activation, pregnancy and HFD-induced obesity, under all of which we measured plasma Gluc activity. As the *in vivo* cell proliferation monitoring system in the present study is based on Gluc being secreted into the blood according to the proliferation rates, with regard especially to β -cells, we were concerned as to whether Gluc secretion would be affected by enhanced insulin secretion. For the revised manuscript, therefore, we further performed glucose tolerance tests on L-MEK mice on day 2 after adenoviral

administration, when β -cell proliferation is markedly enhanced, followed by monitoring both the insulin level and luciferase activity in plasma samples collected from these mice. Thirty minutes after the glucose administrations, both blood glucose and plasma insulin levels were significantly increased as compared to those before glucose loading. In contrast, plasma luciferase activity showed no alterations after glucose loading. Thus, insulin secretion is unlikely to have an impact on the Gluc secretory process in β -cells even when insulin secretion is enhanced. These findings are now presented in Supplemental Figures 5A to 5C and are described in the Results section of the revised manuscript (page 8, lines 18 to 26).

In Figure 3, authors monitored Gluc reporter over time. It would be important to correlate the Gluc signal with a typical proliferation assay ex vivo (such as immunostaining and others) to prove that the produced Gluc signal is indeed in response to cell proliferation. this comment goes to all in vivo models used in this study.

We appreciate the reviewer's important comments. First, to assess whether plasma Gluc activities actually reflect hepatocyte proliferative status *in vivo*, we determined the proportions of Ki67-positive hepatocytes histologically along with evaluating plasma Gluc activities on days 0, 2 and 9 after partial hepatectomy (PHx) using liver-specific Ki67p-Gluc mice (iLKi67p-Gluc mice). Plasma Gluc activities were markedly increased on day 2 followed by reversal to the basal level on day 9 after PHx, which is consistent with the results shown in the original manuscript. In accordance with these results, proportions of Ki67-positive hepatocytes were markedly increased on day 2, followed by reversal to the basal level on day 9 after PHx. In addition, when we compared proportions of Ki67-positive hepatocytes with plasma Gluc activities, these parameters showed a marked and significant correlation ($r = 0.904$, $P < 0.0001$). These results indicate that plasma Gluc activities in iLKi67p-Gluc mice reflect the hepatocyte proliferative status after PHx *in vivo*.

As suggested by the reviewer, we performed similar experiments in other *in vivo* models as well. We examined Ki67-positive β -cells histologically along with evaluating plasma Gluc activities on days 0, 2 and 10 after adenoviral administration in β -cell-specific Ki67p-Gluc mice (i β Ki67p-Gluc mice) treated with adenovirus containing the active mutant gene of MEK-1 (L-MEK mice). Plasma Gluc activities were markedly increased on day 2 followed by reversal to the basal level on day 10 after adenoviral administration. In accordance with these results, proportions of Ki67-positive β -cells in L-MEK mice were markedly increased on day 2 followed by significant decrements on day 10 after

adenoviral administration. When we compared proportions of Ki67-positive β -cells with plasma Gluc activities in these experiments, these parameters showed a marked and very significant correlation ($r = 0.835$, $P < 0.001$).

In a pregnant model as well, both plasma Gluc activities and Ki67-positive β -cells were significantly increased on day 12, followed by significant decrements in both of these parameters on day 21, of pregnancy. Again, proportions of Ki67-positive β -cells and plasma Gluc activities showed a significant correlation in this murine model ($r=0.812$, $p<0.005$).

We performed similar sets of experiments in a high fat diet (HFD)-induced obesity model as well. In this model, increases in plasma Gluc activities were mild, and there was a significant difference between the day 0 and 8 weeks after starting HFD timepoints in the original manuscript. Therefore, we examined Gluc activities and Ki67-positive β -cells at these two timepoints. As expected, both Gluc activities and Ki67-positive β -cells were significantly increased 8 weeks after starting the HFD. Again, proportions of Ki67-positive β -cells and plasma Gluc activities showed a marked and significant correlation ($r = 0.914$, $P < 0.005$).

These findings are now presented in Figures 2B, 2C, 3C, 3D, 4C, 4D, 5C and 5D, and are described in the Results section of the revised manuscript (page 6, lines 26 to 31, page 8, lines 4 to 8, page 9, lines 6 to 11, page 9, lines 23 to 28).

As to the juvenile mouse model and the diurnal rhythm experiments, expression levels of *MKi67* in isolated islets from mice at 4 and 8 weeks of age (juvenile model) and those at ZT8, 16 and 0 (diurnal rhythm experiment) were shown in Supplemental Figures 4A and 4B of the original manuscript, and are now presented as Supplemental figures 7A and 7B of the revised manuscript.

Thus, in all models employed in this study, plasma Gluc activities were confirmed to quantitatively reflect cellular proliferation status as evaluated by Ki67 positivity.

For in culture work, both figure 1 and figure 5, cell proliferation should be monitored by different assays and correlated to Gluc signal

We appreciate the reviewer's important comments. In Figure 1, to determine whether Gluc activity in culture media actually reflects the cell proliferation status, we examined *MKi67* gene expressions of Hepal-6 cells which had been treated with cell cycle inhibitors along with evaluating Gluc activities. *MKi67* gene expressions showed no alterations in response to Cre-adenovirus in Hepal-6 cells, indicating that adenoviral infection does not affect the basal cell proliferation status of Hepal-6 cells. On the other

hand, three types of cell cycle inhibitors, Mitomycin C, Rapamycin and Roscovitine, markedly decreased expressions of *MKi67* in Hepa1-6 cells, observations consistent with the Gluc activities detected in the culture media. In addition, when we compared *MKi67* gene expressions of cells with Gluc activities in culture media in these experiments, these two parameters showed a marked and very significant correlation ($r = 0.903$, $P < 0.0001$). These findings indicate that Gluc activities in culture media reflect the cell proliferation status of Hepa1-6 cells.

With regard to Figure 5, according to the Reviewer's suggestion, we analyzed the correlation between Gluc activities in culture media and *Mki67* gene expression in isolated islets at several glucose concentrations, 5.5, 15 and 25 mM. We found that Gluc activities in culture media and *MKi67* expressions showed a significant correlation ($r = 0.766$, $P < 0.001$). Thus, Gluc activities in culture media reflect the cell proliferation status of isolated islet cells under the high glucose condition.

These findings are now presented in Supplemental Figures 2D to 2G and 6C and are described in the Results section of the revised manuscript (page 5, lines 32 to 36 and page 10, lines 5 to 8).

Responses to reviewer 3

General Comments:

This work describes generation of a novel mouse model that has potential to be adapted by beta cell field in the future and provide methodological advantages to existing methods. Considering these advantages, there are 2 general concerns that attenuate my enthusiasms for this manuscript. Firstly, the manuscript doesn't provide any novel information related to the regulation of beta cell proliferation. The study confirms that beta cell proliferation in mice is induced in response to obesity, pregnancy, etc, but this information has been known for decades. Authors suggest that circadian regulation of beta cell proliferation is a novel observation, however this has been reported by others using standard techniques and much more rigorous mechanistic approaches (e.g. Genes Dev 34, 1650-1665 (2020)). Secondly, the rate of beta cell proliferation in human beta cells is characteristically low and in contrast to mouse models, human beta cells do not appear to augment proliferative rate in response to common metabolic perturbations (e.g. obesity and pregnancy). Thus, the use of mouse models to study mechanisms regulating beta cell proliferation in obesity and diabetes has long been questioned.

With regard to the first point raised by the reviewer, in the previous paper which the reviewer commented on, diurnal variation of β -cell proliferation was observed after ablating β -cells employing DTA, i.e., under condition in which high β -cell proliferation was non-physiologically induced. Therefore, nothing was revealed about diurnal variation of physiological β -cell proliferation. We were able to observe diurnal variation of β -cell proliferation during the juvenile period, i.e., under physiological conditions. Our sensitive *in vivo* cell proliferation monitoring system, for the first time, allowed us to detect subtle but physiological diurnal variation of β -cell proliferation without being affected by individual differences.

With regard to the second point, we agree with the reviewer's comments that β -cell proliferation capacity is lower in humans than in mice. However, several studies found β -cell mass to be larger in obese than in lean subjects, indicating low but adequate proliferation capacity in humans as well. Therefore, activating this capacity may lead to a curative therapy for diabetes. However, the human islet cell supply is insufficient and individual differences in human islet cells remain a challenge when screening for β -cell proliferative factors. In this regard, screening β -cell proliferative factors using islet cells isolated from our β -cell-specific Ki67p-Gluc mice and administering the discovered factors to these mice might hold promise as an efficient strategy.

More importantly, we would like to emphasize that the most important point of the present study is the development of a novel strategy for monitoring real-time cell proliferation *in vivo* in the same individuals. Since our Ki67p-Gluc mouse system, as reported herein, is highly sensitive, allowing quantitative detection of proliferation of very small amounts of cells, e.g., pancreatic β -cells, this system may facilitate unraveling the time courses of a variety of *in vivo* phenomena. In addition, this system may apply to screening for proliferative factors designed to increase the masses of a variety of cells, such as not only β -cells for treating diabetes, but also hepatocytes for liver dysfunction, endocrine cells for various deficiencies, cardiomyocytes for cardiac failure, myocytes for sarcopenia, vascular cells for circulation insufficiencies, and so on, by crossing Ki67p-LSL-Gluc mice with different promoter-driven Cre expressing mouse lines. In addition, this system may be useful when searching for factors suppressing cellular proliferation, with the aim of developing anti-cancer therapies. Thus, these novel strategies may contribute to major advancements in many areas of biological and medical research including those of development, growth, tissue repair, organ regeneration and cancer. To enhance the understanding of our aim and the significance of this study, we have rewritten the Introduction (page 4, lines 4 to 19) and Discussion sections (page 14, lines 16 to 34) of the revised manuscript.

Specific Comments:

1. Increase in luciferase activity (as an indicator of beta cell proliferation) has to be validated by objective measures of beta cell proliferation (e.g. protein expression of Ki67, PCNA or BrdU by immunodetection methods). The existing immunodetection methods (e.g. Ki67 staining by immunohistochemistry) allows for quantification of the frequency or the rate of beta cell proliferation (e.g. % proliferating beta cells). Correspondingly, existing methodology has shown a several fold increase in the frequency of beta cell proliferation in response to obesity/pregnancy. It will be important to compare/contrast how an increase in the frequency of beta cell proliferation compares to fold induction of luciferase activity presented in the current study.

As suggested, we determined whether plasma Gluc activities actually reflect the β -cell proliferative status *in vivo*. We first examined Ki67-positive β -cells histologically along with evaluating plasma Gluc activities on days 0, 2 and 10 after adenoviral administration in β -cell-specific Ki67p-Gluc mice (i β Ki67p-Gluc mice) treated with adenovirus containing the active mutant gene of MEK-1 (L-MEK mice). Plasma Gluc activities and proportions of Ki67-positive β -cells were similarly increased on day 2, followed by reversal to the basal level on day 10 after adenoviral administration. In addition, proportions of Ki67-positive β -cells and plasma Gluc activities showed a marked correlation with a very low p-value ($r = 0.835$, $P < 0.001$).

Next, we performed similar experiments in a pregnant model. Both plasma Gluc activities and Ki67-positive β -cells were significantly increased on day 12, followed by significant decrements on day 21, of pregnancy. In addition, the proportion of Ki67-positive β -cells and plasma Gluc activities showed a marked and significant correlation ($r = 0.812$, $P < 0.005$).

In addition, we performed similar sets of experiments in a high fat diet (HFD)-induced obesity model. In this model, increases in plasma Gluc activities were mild, and a significant difference was detected between day 0 and 8 weeks after starting HFD in the original manuscript. Therefore, we examined Gluc activities and Ki67-positive β -cells at these two timepoints. As expected, both Gluc activities and Ki67-positive β -cells were significantly increased 8 weeks after starting the HFD. Again, the proportion of Ki67-positive β -cells and plasma Gluc activities showed a strong and significant correlation ($r = 0.914$, $P < 0.005$).

These results clearly indicate that plasma Gluc activities of i β Ki67p-Gluc mice reflect the

β -cell proliferative status in all models employed in the present study.

These findings are now presented in Figures 3C, 3D, 4C, 4D, 5C and 5D and are described in the Results section of the revised manuscript (page 8, lines 4 to 8, page 9, lines 6 to 11, page 9, lines 23 to 28).

2. Beta cell secretory pathway is altered in response to conditions associated with obesity and Type 2 diabetes. This is related to commonly associated abnormalities in beta cell ER function, protein folding, autophagy, etc. Thus, authors should confirm that expected changes in ER and secretory functions in “stressed/diabetic” beta cells will not interfere with Gluc production/secretion and thus result in inaccurate approximation of beta cell proliferation using this method.

We appreciate the reviewer’s important and insightful comments. As suggested, we examined whether ER stress in β -cells affected Gluc secretion from islets isolated from $i\beta$ Ki67p-Gluc mice. High glucose conditions not only promote β -cell proliferation but also induce ER stress (Lipson KL, et al, *Cell Metab*, 2006, Elouil H, et al, *Diabetologia*, 2007). Therefore, we treated isolated islets from $i\beta$ Ki67p-Gluc mice under conditions of low (5.5 mM) and high (15 mM and 25 mM) glucose, followed by evaluating both ER stress and cellular proliferation. As expected, expressions of *Bip*, an ER stress marker, were dose-dependently and significantly increased along with elevation of glucose concentrations in culture media, indicating that ER stress in islet cells was indeed induced by high glucose conditions. Under these experimental conditions, we examined both Gluc activities in culture media and cellular *Mki67* expressions. Notably, *Mki67* expressions in islet cells and Gluc activities in culture media showed a significant correlation, with a very low *p* value ($r = 0.766$, $P < 0.001$). These results demonstrate that effects of ER stress were minimal on Gluc secretion from β -cells of $i\beta$ Ki67p-Gluc mice. These findings are now presented in Supplemental Figures 6A to 6C and are described in the Results section of the revised manuscript (page 9, line 32 to page 10, line 9), and appropriate new references are cited as Refs 30 to 35.

3. Authors should also provide data on glucose and KCL-stimulated insulin secretion as well as insulin content in isolated islets of $i\beta$ Ki67pGluc mice. This data should be presented and compared to proper controls (e.g. RIP-Cre and floxed mice) and is important to make sure there are no alterations in beta cell secretory function in this model.

As suggested, we examined high glucose (HG)- and KCl-stimulated insulin secretions of islets isolated from $i\beta Ki67pGluc$ mice and compared the values with those from two types of controls, RIP-Cre and floxed mice. Both HG and KCl significantly stimulated insulin secretions from $i\beta Ki67pGluc$ mouse islets, and insulin secretory responses were comparable to those from RIP-Cre- and floxed mice. In addition, insulin contents of isolated islets differed minimally among $i\beta Ki67pGluc$, RIP-Cre and floxed mice. Thus, it is unlikely that β -cell $Ki67pGluc$ altered β -cell secretory function. These findings are now presented in Supplemental Figures 4A and 4B and are described in the Results section of the revised manuscript (page 8, lines 12 to 18).

Reviewers' Comments:

Reviewer #1:

Remarks to the Author:

This revised manuscript, by Hiroto Sugawara and colleagues, is significantly improved compared to the first version since they perform controls for Ki-67 expression (a standard readout of cell proliferation) and GLuc detection in blood or culture medium. There remain several issues that need addressing, however, before I feel that it could be published. The main problem is the exclusive use of Ki-67 expression in the same cells as an "independent" marker of cell proliferation to control for the latter, because it is not independent. Although this is useful because it ensures that differences in GLuc expression do reflect differences in Ki-67 expression in the same cells, both in vivo and in vitro, which was not the case in the initial manuscript, this is nevertheless slightly circular: while it is reasonable to assume that Ki-67 expression reflects cell proliferation in different circumstances, this requires verification by an independent marker of cell proliferation – in other words, not Ki-67. It was shown in vivo and in vitro by Sobecki et al., 2016, 2017, Miller et al 2018 and Mrouj et al. 2021, that Ki-67 expression can be uncoupled from cell proliferation in both directions. Cells can proliferate in the absence of Ki-67 (siRNA, shRNA or gene ablation), while situations that abrogate Ki-67 degradation allow Ki-67 expression in non-proliferating cells. It is important to mention this caveat. And both Sobecki et al., 2017 and Miller et al., 2018, found that Ki-67 protein is not an all-or-nothing marker of cell proliferation and low levels are detected in cells having recently exited the cell cycle. Nevertheless, Sobecki et al., 2017 showed that treatments expected to affect cell proliferation but which also affect Ki-67 transcription directly (eg CDK4/6 inhibition) affect Ki-67 levels and cell proliferation (as assessed by independent readouts) similarly. Such independent readouts are critical controls, and they work well and are perfectly feasible.

The second issue is that GLuc is expected to reflect Ki-67 mRNA levels as both are controlled by the same promoter in the same cells, yet Ki-67 mRNA levels may not be proportional to cell proliferation as Ki-67 is controlled at the protein level. This appears to be reflected in their new data. Thus, while it is reassuring that Ki-67 promoter activity controls Ki-67 mRNA and GLuc activity similarly (though not necessarily proportionally – see below) , to ensure that this accurately reflects cell proliferation, another control is required. Specifically, the authors have performed immunohistochemistry for Ki-67 to quantify the fraction of proliferating cells in experiments shown in Figure 2. This is better than measuring Ki-67 mRNA levels, as shown in supplemental Fig 1 for controls for fig 1, because mRNA levels are only part of the control of Ki-67 protein expression, which is also regulated by degradation. Thus, in Fig 1, GLuc is five-fold lower in medium from MIN6 cells compared to Hepa1-6 cells, yet in Fig S1, Ki-67 mRNA levels are only two-fold lower. This is not a good correlation. Since these are in vitro experiments, it is very easy to quantify for the percentage of proliferating cells, and it would be important to know whether this is two-fold or five-fold lower. This could be done by quantifying the percentage of Ki-67 protein-positive cells as well as independent markers, eg phospho histone H3 (mitotic marker), mitotic index, BrdU incorporation, PCNA-positive nuclei, cyclin A2 positive nuclei, etc. Only then will they know how a change in GLuc relates to a change in cell proliferation. Similar controls should also be performed on samples from in vivo experiments as this material is available.

If such controls were done, I would be much more favourably inclined towards this manuscript.

Reviewer #2:

Remarks to the Author:

The authors have done a wonderful job addressing previous critiques and the manuscript is much more improved.

Reviewer #3:

Remarks to the Author:

I have no further questions for the authors.

Responses to Reviewer 1

Reviewer #1 (Remarks to the Author):

The main problem is the exclusive use of Ki-67 expression in the same cells as an "independent" marker of cell proliferation to control for the latter, because it is not independent. Although this is useful because it ensures that differences in GLuc expression do reflect differences in Ki-67 expression in the same cells, both in vivo and in vitro, which was not the case in the initial manuscript, this is nevertheless slightly circular: while it is reasonable to assume that Ki-67 expression reflects cell proliferation in different circumstances, this requires verification by an independent marker of cell proliferation – in other words, not Ki-67. It was shown in vivo and in vitro by Sobecki et al., 2016, 2017, Miller et al 2018 and Mrouj et al. 2021, that Ki-67 expression can be uncoupled from cell proliferation in both directions. Cells can proliferate in the absence of Ki-67 (siRNA, shRNA or gene ablation), while situations that abrogate Ki-67 degradation allow Ki-67 expression in non-proliferating cells. It is important to mention this caveat. And both Sobecki et al., 2017 and Miller et al., 2018, found that Ki-67 protein is not an all-or-nothing marker of cell proliferation and low levels are detected in cells having recently exited the cell cycle. Nevertheless, Sobecki et al., 2017 showed that treatments expected to affect cell proliferation but which also affect Ki-67 transcription directly (eg CDK4/6 inhibition) affect Ki-67 levels and cell proliferation (as assessed by independent readouts) similarly. Such independent readouts are critical controls, and they work well and are perfectly feasible.

We appreciate the reviewer's insightful comments. We agree that another control, independent of Ki-67 expression, is crucial for evaluating the usefulness of plasma Gluc activity. Therefore, we additionally performed histological analyses examining proportions of cells positive for phospho-histone H3 (PHH3), a mitotic marker other than Ki67, along with evaluating plasma Gluc activities in *in vivo* studies.

First, we analyzed proportions of PHH3-positive hepatocytes on days 0, 2 and 9 after partial hepatectomy (PHx) using liver-specific Ki67p-Gluc mice (iLK67p-Gluc mice). Plasma Gluc activities were markedly increased on day 2, followed by return to the basal level by day 9 after PHx. Consistent with these results, ratios of PHH3-positive cells to all hepatocytes were markedly increased on day 2 followed by return to the basal level by day 9 after PHx. In addition, when we compared proportions of PHH3-positive hepatocytes in the liver and plasma Gluc activities in these experiments, these two

parameters showed a marked and significant correlation ($r = 0.974$, $P < 0.0000001$). These findings using hepatocytes confirmed our conclusion that plasma Gluc activities in iLKi67p-Gluc mice quantitatively reflect the proliferative status after PHx *in vivo* which was evaluated employing two different cell proliferation markers, Ki67 and PHH3.

Next, we examined PHH3-positive β -cells histologically along with evaluating plasma Gluc activities on days 0, 2 and 10 after adenoviral administration in β -cell-specific Ki67p-Gluc mice (i β Ki67p-Gluc mice) treated with adenovirus containing the active mutant gene of MEK-1. Ratios of PHH3-positive β -cells and plasma Gluc activities were both markedly increased on day 2, followed by reversal to the basal level on day 10 after adenoviral administration. In addition, proportions of PHH3-positive β -cells and plasma Gluc activities showed a marked and significant correlation ($r = 0.823$, $P < 0.005$). We further performed similar experiments in a pregnant model. Plasma Gluc activities and PHH3-positive β -cells were both significantly increased on day 12, followed by significant decrements on day 21, of pregnancy. In addition, proportions of PHH3-positive β -cells and plasma Gluc activities showed a significant correlation ($r = R = 0.777$, $P < 0.005$). Moreover, we performed similar sets of experiments in a high fat diet (HFD)-induced obesity model. Again, proportions of PHH3-positive β -cells and plasma Gluc activities showed a significant correlation ($R = 0.749$, $P < 0.05$). These results clearly demonstrate that, in not only hepatocytes but also pancreatic β -cells, plasma Gluc activities reflect the cellular proliferative status *in vivo* in all models employed in the present study.

These findings are now presented in Supplemental Figures 4A, 4B, 5A, 5B, 8A, 8B, 9A and 9B, and described in the Results section of the revised manuscript (page 6, lines 28 to 35, page 8, lines 8 to 14, page 9, lines 12 to 18 and page 9, line 31 to page 10, line 1).

The second issue is that GLuc is expected to reflect Ki-67 mRNA levels as both are controlled by the same promoter in the same cells, yet Ki-67 mRNA levels may not be proportional to cell proliferation as Ki-67 is controlled at the protein level. This appears to be reflected in their new data. Thus, while it is reassuring that Ki-67 promoter activity controls Ki-67 mRNA and GLuc activity similarly (though not necessarily proportionally – see below), to ensure that this accurately reflects cell proliferation, another control is required. Specifically, the authors have performed immunohistochemistry for Ki-67 to quantify the fraction of proliferating cells in experiments shown in Figure 2. This is better than measuring Ki-67 mRNA levels, as shown in supplemental Fig 1 for controls for fig 1, because mRNA levels are only part of the control of Ki-67 protein expression, which is also regulated by degradation. Thus, in Fig 1, GLuc is five-fold lower in medium from

MIN6 cells compared to Hepa1-6 cells, yet in Fig S1, Ki-67 mRNA levels are only two-fold lower. This is not a good correlation. Since these are in vitro experiments, it is very easy to quantify for the percentage of proliferating cells, and it would be important to know whether this is two-fold or five-fold lower. This could be done by quantifying the percentage of Ki-67 protein-positive cells as well as independent markers, eg phospho histone H3 (mitotic marker), mitotic index, BrdU incorporation, PCNA-positive nuclei, cyclin A2 positive nuclei, etc. Only then will they know how a change in GLuc relates to a change in cell proliferation. Similar controls should also be performed on samples from in vivo experiments as this material is available.

We appreciate the reviewer's important comments. Again, we agree with the comment that another control, independent of Ki-67 expression, is crucial for evaluating the usefulness of plasma Gluc activity. As noted in the aforementioned responses, we histologically examined cellular proliferation status *in vivo* by evaluating PHH3 expression. In addition, we analyzed proportions of cells positive for PHH3, in both the Hepa1-6 and the MIN6 cell lines, *in vitro*. In these experiments, we obtained results showing that cells positive for either Ki67 or PHH3 protein were consistently higher, by 2-fold, in Hepa1-6 than in MIN6 cells, indicating higher cell proliferation activity in the former. These findings are now presented in Supplemental Figures 1C and 1D, and described in the Results section of the revised manuscript (page 5, lines 23 to 27).

As the reviewer pointed out, Gluc is five-fold lower in medium from MIN6 than Hepa1-6 cells. Therefore, it is possible that amounts of Gluc produced by PHH3-positive cells vary between hepatocytes and β -cells. However, as noted in the aforementioned responses, plasma Gluc activities showed marked and significant correlations with cellular proliferation states of both hepatocytes and β -cells, thereby reflecting the cellular proliferative status in all models employed in this study. Given that the actual application of this system is a time series comparison under various conditions in the same murine models or cells, the differences in Gluc production among cell types do not diminish the usefulness of this system for monitoring real time proliferations of targeted cell types.

Reviewers' Comments:

Reviewer #1:

Remarks to the Author:

Review of NCOMMS-21-20580B, Sugawara et al.

The authors seem to have done a good job in adding the additional controls that were needed; nevertheless it seems that the new data are not always well presented. In particular, in Supplementary Figure 5, the increase in phospho-histone H3 (PHH3) staining in days 2 and, to a lesser extent, day 10, shown in the left panel, is not illustrated by the microscopy data shown in the right panel: two of the three white arrowheads at day 2 and one of the two arrowheads at day 10 appear to be pointing to PHH3-negative cells, whereas the legend states that they are all positive. There is a similar problem with supplementary figure 8A, microscopy panel, day 12 – I can see at most only one PHH3 positive cell out of the four white arrowheads. Is it possible that the original PHH3 staining is obscured by the merge of the different fluorescent channels, or have these been scored incorrectly? Could the authors show the unmerged PHH3 staining for these figures? Assuming that there is no problem with this data, I would be happy to accept the paper, on the condition that the grammatical errors in the title are corrected. As is already the case in the rest of the manuscript, real-time should be hyphenated as it is an adjective, and proliferation should not have an "s" as it is never plural.

Responses to Reviewer 1

Reviewer #1 (Remarks to the Author):

The authors seem to have done a good job in adding the additional controls that were needed; nevertheless it seems that the new data are not always well presented. In particular, in Supplementary Figure 5, the increase in phospho-histone H3 (PHH3) staining in days 2 and, to a lesser extent, day 10, shown in the left panel, is not illustrated by the microscopy data shown in the right panel: two of the three white arrowheads at day 2 and one of the two arrowheads at day 10 appear to be pointing to PHH3-negative cells, whereas the legend states that they are all positive. There is a similar problem with supplementary figure 8A, microscopy panel, day 12 – I can see at most only one PHH3 positive cell out of the four white arrowheads. Is it possible that the original PHH3 staining is obscured by the merge of the different fluorescent channels, or have these been scored incorrectly? Could the authors show the unmerged PHH3 staining for these figures?

We apologize for presenting unclear images of PHH3 staining. As suggested, we replaced the PHH3 staining images in Supplemental Figures 5 and 8 with images showing PHH3-positive cells more clearly. In the original images, PHH3 staining was obscured because the fluorescence signals of DAPI (blue) were strong while the fluorescence signal of PHH3 (red) was weak. Therefore, we processed the images in the original manuscript by attenuating DAPI staining while enhancing PHH3 staining. The same processing was performed on all images in both supplementary Figure 5 (days 0, 2 and 10) and 8 (days 0, 12 and 21). In addition, as requested by the reviewer, unmerged images of supplementary Figures 5 and 8 of the revised manuscript are shown below, including single staining results obtained with PHH3, insulin or DAPI, double (insulin and PHH3) staining without DAPI, and triple staining (insulin, PHH3 and DAPI). Since non-specific signals were commonly observed in both insulin and PHH3 single stained images, we first excluded yellow areas in the double (insulin and PHH3) staining images. A previous report showed that PHH3 appears as small nuclear domains and spreads throughout the nucleus during mitosis (Hendzel et al, *Chromosoma*, 106: 348-360, 1997). Therefore, we judged PHH3-positive β cell nuclei according to the following criteria: 1) overlapping with DAPI staining 2) the cytosol being positive for insulin and 3) speckled (blue arrowheads) or homogeneous (pink arrowheads) staining pattern which is characteristic of PHH3 staining. Images for double (insulin and PHH3) staining clearly showed that β -cells indicated by arrowheads are positive for PHH3 and that triple staining results

revealed the staining area to be within the nuclei because of DAPI positivity.

Images of Supplementary Figure 5

Images of staining with PHH3 alone in the revised supplementary Figure 5

Images of staining with insulin alone in the revised supplementary Figure 5

Images of staining with DAPI alone in the revised supplementary Figure 5

Images of staining without DAPI in the revised supplementary Figure 5

Images of triple staining in the revised supplementary Figure 5

Blue arrowheads: Nuclei with the speckled staining pattern

Pink arrowheads: Nuclei with the homogenous staining pattern

Images of Supplementary Figure 8

Images of staining with PHH3 alone in the revised supplementary Figure 8

Images of staining with insulin alone in the revised supplementary Figure 8

Images of staining with DAPI alone in the revised supplementary Figure 8

Images of staining without DAPI in the revised supplementary Figure 8

Images of triple staining in the revised supplementary Figure 8

Blue arrowheads: Nuclei with the speckled staining pattern

Pink arrowheads: Nuclei with the homogenous staining pattern

Assuming that there is no problem with this data, I would be happy to accept the paper, on the condition that the grammatical errors in the title are corrected. As is already the case in the rest of the manuscript, real-time should be hyphenated as it is an adjective, and proliferation should not have an “s” as it is never plural.

We appreciate the reviewer’s comments. We corrected the grammatical errors throughout the manuscript, including the title, pointed out by the reviewer (“real-time” and “proliferation”).